# POSTERCRAFT: RETHINKING HIGH-QUALITY AESTHETIC POSTER GENERATION IN A UNIFIED FRAMEWORK

**Sixiang Chen**[1,2]\*, **Jianyu Lai**[1]\*, **Jialin Gao**[2]\*, **Tian Ye**[1], **Haoyu Chen**[1], **Hengyu Shi**[2],
**Shitong Shao**[1], **Yunlong Lin**[3], **Song Fei**[1], **Zhaohu Xing**[1], **Yeying Jin**[4],
**Junfeng Luo**[2], **Xiaoming Wei**[2], **Lei Zhu**[1,5]†

[1]The Hong Kong University of Science and Technology (Guangzhou)
[2]Meituan
[3]Xiamen University
[4]National University of Singapore
[5]The Hong Kong University of Science and Technology

## ABSTRACT

Generating aesthetic posters is more challenging than simple design images: it requires not only precise text rendering but also the seamless integration of abstract artistic content, striking layouts, and overall stylistic harmony. To address this, we propose PosterCraft, a unified framework that abandons prior modular pipelines and rigid, predefined layouts, allowing the model to freely explore coherent, visually compelling compositions. PosterCraft employs a carefully designed, cascaded workflow to optimize the generation of high-aesthetic posters: (i) large-scale text-rendering optimization on our newly introduced Text-Render-2M dataset; (ii) region-aware supervised fine-tuning on HQ-Poster-100K; (iii) aesthetic-text reinforcement learning via best-of-n preference optimization; and (iv) joint vision–language feedback refinement. Each stage is supported by a fully automated data-construction pipeline tailored to its specific needs, enabling robust training without complex architectural modifications. Evaluated on multiple experiments, PosterCraft significantly outperforms open-source baselines in rendering accuracy, layout coherence, and overall visual appeal—approaching the quality of SOTA commercial systems.

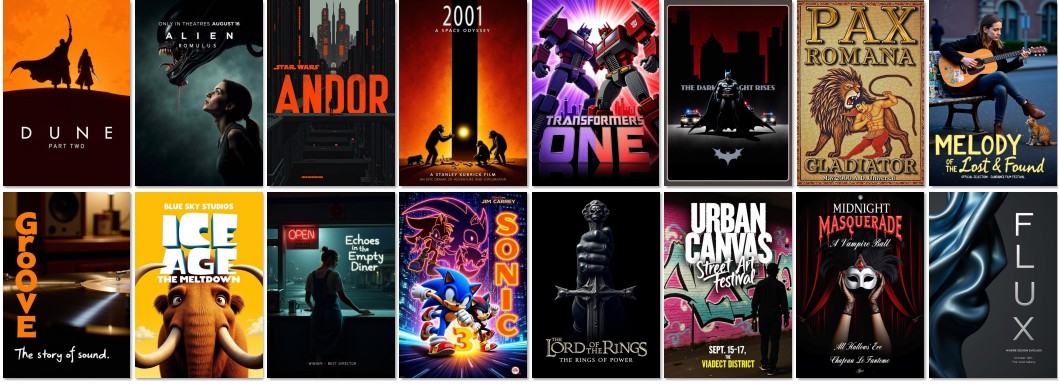

Figure 1: **Aesthetic posters generated by PosterCraft** demonstrate that backgrounds, layouts, and typographic designs are produced directly from textual input without modular designs, highlighting its unified framework for visually consistent and appealing designs.

---

\*Equal contribution.
†Corresponding author.

# 1 INTRODUCTION

Despite recent advances in automated visual design, aesthetic poster generation remains a formidable challenge and is still relatively underexplored. Existing generative approaches primarily focus on foundational tasks such as text rendering Chen et al. (2023b); Liu et al. (2024); Tuo et al. (2023); Chen et al. (2023a) or the creation of specific product-oriented posters Wang et al. (2025); Gao et al. (2025); Podell et al. (2023), offering limited capacity to produce high-quality, aesthetically compelling outputs. These methods often fall short of addressing the multifaceted demands of aesthetic poster design, which requires not only *(i)* accurate and stylistically coherent text, but also *(ii)* the creation of abstract and visually appealing artistic content and *(iii)* the striking layouts and holistic stylistic consistency. Therefore, aesthetic poster generation demands a more comprehensive synthesis of content, form, and communicative intent.

Recent approaches to aesthetic poster generation Chen et al. (2025a); Yang et al. (2024b); Seol et al. (2024); Tang et al. (2023) have primarily followed a modular design paradigm. Typically, a fine-tuned vision-language model (VLM) acts as a layout planner, suggesting text content and positioning. The suggestions are then overlaid onto a separately generated background, or used as hard constraints for the generative model to follow. However, this design strategy presents several limitations. *(i) Lack of aesthetic consistency:* it undermines the visual and stylistic coherence essential for aesthetic poster creation. *(ii) Limited visual quality:* it constrains the upper bound of visual quality due to the decoupled design process and heavy reliance on the VLM's accuracy and robustness. In contrast, existing end-to-end design-centered generation approaches Chen et al. (2023b); Inoue et al. (2024); Wang et al. (2025); Gao et al. (2025) remain limited to relatively simple tasks, such as greeting cards or product compositions, which lack the visual and structural complexity of high-quality aesthetic posters. Additionally, while powerful foundation models https://github.com/black-forest labs/flux (2024); Esser et al. (2024); AI (2024) have demonstrated impressive capabilities in generating complex natural images, they still fall short of meeting all the specific requirements of aesthetic posters. (e.g. precise text rendering, abstract artistic content, and holistic stylistic coherence). For this, we classify it as *(iii) Simplified use cases*. More importantly, the absence of large-scale, versatile datasets tailored specifically for aesthetic poster generation has further constrained the development of fully generative solutions—*(iv) Absence of targeted datasets*.

To move beyond the limitations of modular and simply scoped designs, we leverage foundation models to explore unified generation for aesthetic posters, aiming for visually coherent and compelling results. We argue that incremental, component-level improvements are insufficient for major aesthetic gains. Instead, we propose a unified framework, *PosterCraft*, which includes a comprehensive workflow to systematically perform four critical stages: *(i)* scalable text rendering optimization, *(ii)* high-quality poster fine-tuning, *(iii)* aesthetic-text reinforcement learning, and *(iv)* vision-language feedback refinement. We construct specialized datasets for each stage through automated pipelines, enabling robust training and future research in aesthetic poster generation. This framework empowers the trained model to generate high-quality posters end-to-end. Experiments show our approach outperforms existing baselines and is competitive with several closed-source models.

Overall, our contributions can be summarized as follows:

- **A unified framework for aesthetic poster generation:** We revisit aesthetic poster generation through an end-to-end approach for high-quality, visually coherent posters, surpassing prior modular pipelines and methods focused on simpler or product-centric designs.

- **A cascade workflow for high-quality poster optimization:** We propose a unified training pipeline with four stages: (i) scalable text rendering optimization, (ii) high-quality poster fine-tuning, (iii) aesthetic-text reinforcement learning, and (iv) vision-language feedback refinement. Each stage targets a key challenge in aesthetic poster generation, enabling the model to produce artistically compelling results at inference time.

- **Stage-specific, fully automated dataset construction:** We construct specialized datasets for each workflow stage using automated collection and filtering, tailored to the unique demands of aesthetic poster generation. These datasets overcome the limitations of resources and support more robust, transferable training.

- **Superior performance over existing baselines:** Extensive experiments show that our method significantly outperforms open-source baselines in terms of aesthetic quality and layout structure, and achieves competitive performance compared to commercial systems.

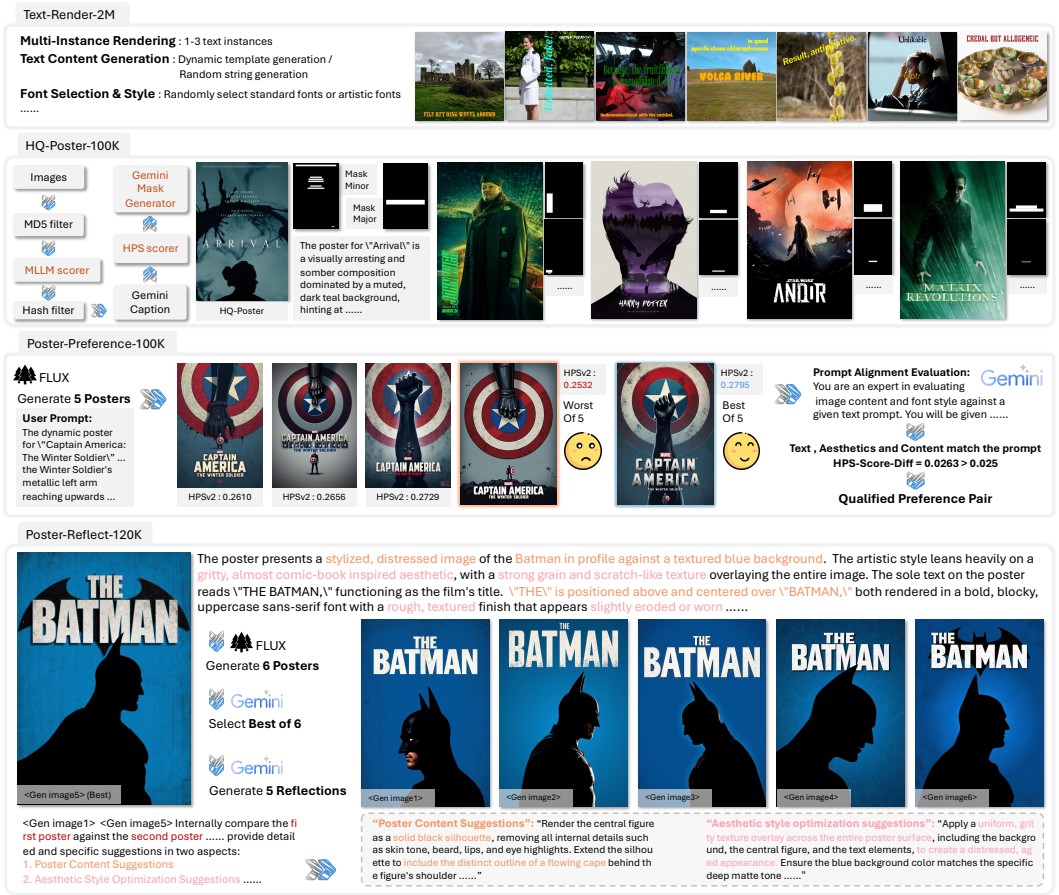

Figure 2: **Four datasets of PosterCraft** across four stages: (i) Text-Render-2M for text rendering optimization, (ii) HQ-Poster-100K, comprising over 100K high-quality posters with masks and captions, (iii) Poster-Preference-100K, yielding 6K high-quality preference pairs from 100K generated samples, (iv) Poster-Reflect-120K, constructing 64K feedback pairs from 120K generated posters.

## 2    RELATED WORKS

**Image Generation for Design Images.** Diffusion models have rapidly matured across a wide range of vision problems beyond generic text-to-image synthesis Chen et al. (2024b); Lin et al. (2025); Fei et al. (2025); Chen et al. (2023c; 2025c); Li et al. (2023). Meanwhile, for high-fidelity visual synthesis, diffusion/flow-matching families remain the dominant backbone in recent systems Chen et al. (2026); Esser et al. (2024); Geng et al. (2025); https://github.com/black-forest labs/flux (2024); Wu et al. (2025); Ye et al. (2025). Recently, design-centric image generation has transitioned from early GAN- and VAE-based generators to more powerful diffusion-based frameworks Chen et al. (2023b); Tuo et al. (2023); Wang et al. (2025); Gao et al. (2025); Inoue et al. (2023); Zheng et al. (2023); Chen et al. (2023a). Notably, LayoutDiffusion Zheng et al. (2023) reformulated layout generation as a discrete token denoising process, achieving significant gains. Advanced pipelines like TextDiffuser Chen et al. (2023b;a) adopted a two-stage approach: a transformer-based layout planner followed by a diffusion model conditioned on OCR-derived masks to generate coherent text images from prompts. DesignDiffusion Wang et al. (2025) improved text-centric design by enhancing prompts with character-level embeddings and applying a localization loss to boost text rendering accuracy. Despite recent progress, many methods imposed rigid pre-layout constraints on text and layout, often undermining overall aesthetic coherence. Furthermore, their focus on specific domains such as product ads or greeting cards also limited the complexity and creativity of the tasks they address. In contrast, our approach adopts a unified framework that integrates text rendering, artistic content, and layout design within a single inference process.

**VLM for Image Generation.** Vision–language models served dual roles as both high-level planners Lin et al. (2023); Luo et al. (2024); Feng et al. (2023); Yang et al. (2024b;a); Chen et al. (2025a);

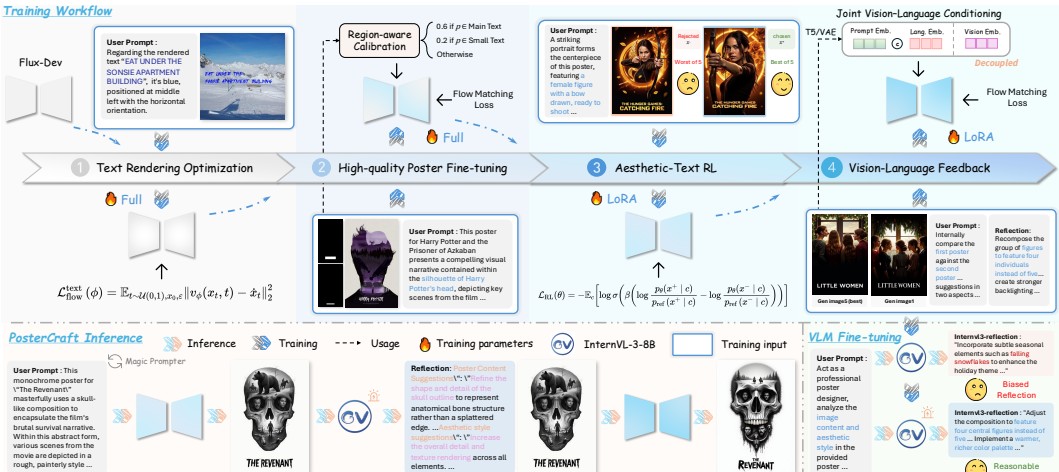

Figure 3: **The PosterCraft pipeline** has four stages: (1) Text Rendering Optimization to improve text accuracy and fidelity; (2) High-Quality Poster Fine-Tuning with region-aware calibration to poster styling across text and non-text regions; (3) Aesthetic-Text Reinforcement Learning to instill detailed aesthetic and content preferences; and (4) Joint Vision–Language Feedback, integrating multimodal reflections for refined outputs. At inference, the fine-tuned model generates high-quality aesthetic posters end-to-end from a single prompt, with an optional VLM-driven critique loop.

Tang et al. (2023) and fully end-to-end generators Zhou et al. (2024a); Xie et al. (2024); Team (2024a); Ma et al. (2024); Zhou et al. (2024a); Wu et al. (2024). Fine-tuned adaptations PosterLlama Seol et al. (2024) and PosterLLava Yang et al. (2024b) employed HTML- or JSON-formatted tokens to produce content-aware graphic designs. POSTA Chen et al. (2025a) further leveraged a fine-tuned VLM as an "aesthetic designer", applying modular design atop existing high-quality images. Unified transformer architectures like TransFusion Zhou et al. (2024a), and JanusFlow Ma et al. (2024) advanced this paradigm by generating image and text tokens in one architecture, enabling one-shot synthesis of complete visual compositions. Complementing these approaches, feedback-driven pipelines Li et al. (2024; 2025) incorporated VLM-based critics to assess object accuracy during training. Nevertheless, these VLM-driven methods continued to struggle with maintaining cohesive aesthetic consistency, and faced an upper bound in layout complexity imposed by the models' architectural capacity and the scope of their limited training data.

## 3 UNIFIED WORKFLOW AND SPECIFIC DATASET

In this work, we rethink aesthetic poster generation: a high-capacity model optimized through a unified workflow can directly produce high-quality, fully rendered posters without modular design. This allows holistic integration of text, visuals, and layout, while leveraging vision-language feedback during inference for greater coherence and aesthetic appeal. Unlike prior methods that rely on layout embeddings Zheng et al. (2023); Gao et al. (2025); Chen et al. (2023b) or external VLM-based designers Chen et al. (2025a); Yang et al. (2024b); Seol et al. (2024) (i.e. inherently restrict a model's expressive freedom), we unlock the potential of a standard diffusion backbone via workflow optimization rather than intricate architectural modifications. Our paradigm remains compatible with existing techniques and provides a flexible foundation for future advances. Fig.3 shows our unified optimization workflow, and Fig.2 the dataset pipeline supporting each stage.

### 3.1 SCALABLE TEXT RENDERING OPTIMIZATION

In the first stage of our workflow, we target the challenge of accurate text rendering, a persistent bottleneck in poster generation. Progress is hindered by two factors: (i) the scarcity of large-scale, high-quality datasets with perfectly rendered text, and (ii) most available text datasets feature plain or low-quality backgrounds, which easily make the model lose the ability to represent common backgrounds. To overcome these issues, we construct Text-Render-2M via an automated pipeline, producing 2 million samples with diverse text (varying in content, size, count, placement, and rotation) rendered crisply onto high-quality backgrounds. Each text instance is paired with precise captions merged seamlessly with existing image captions. This dataset ensures both 100% text

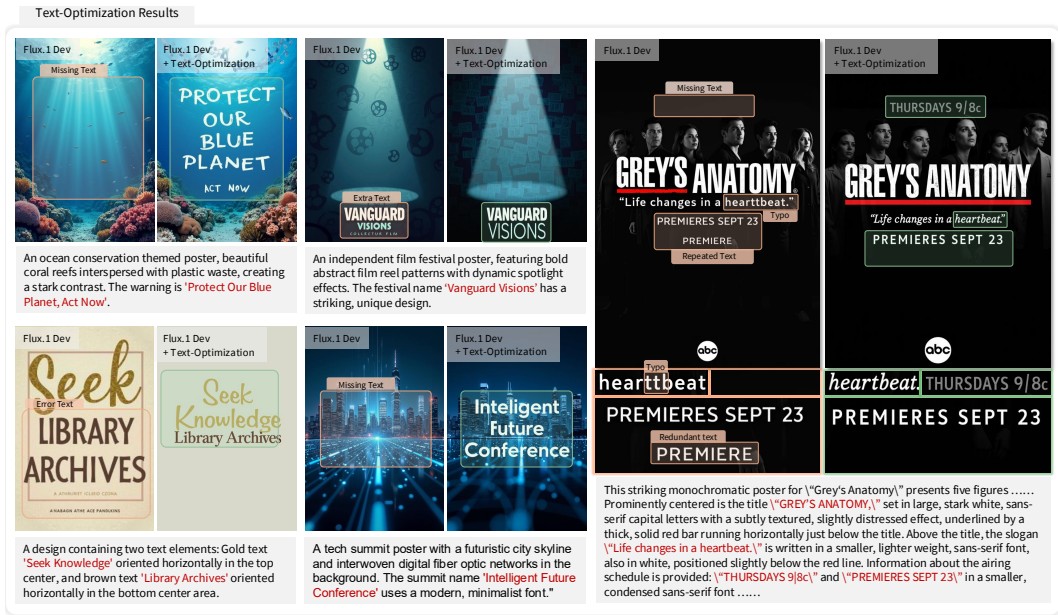

Figure 4: **Comparison of text rendering on poster typography, plain-text scenes, and long-form text posters**. Each pair shows the Flux.1 dev baseline (left), exhibiting missing, repeated, or error text, alongside the optimized output (right) after our scalable text rendering optimization, demonstrating marked gains in text fidelity, alignment, and accuracy.

rendering accuracy and rich background diversity, enhancing fidelity and robustness in real-world scenarios. Fig.2 illustrates Text-Render-2M, with construction details and examples provided in the supplementary. We then fine-tune foundation models on paired Text-Render-2M using the flow matching loss Esser et al. (2024) to enhance text rendering:

$$\mathcal{L}_{\text{flow}}^{\text{text}}(\phi) = \mathbb{E}_{t \sim \mathcal{U}(0,1), \, x_0, \, \varepsilon} \big\| v_\phi(x_t, t) - \dot{x}_t \big\|_2^2, \tag{1}$$

where $x_t = \alpha_t x_0 + \sigma_t \varepsilon$ follows the forward noising schedule, $\dot{x}_t$ is its time derivative, and $v_\phi$ predicts the velocity field. Discretized over timesteps, this loss encourages the model to match the true data flow and yields markedly improved text rendering. As shown in Fig.4, the model augmented with our text-rendering optimization achieves significant gains in both rendering accuracy and text alignment across poster typography, plain-text scenes, and long-text posters.

## 3.2 HIGH-QUALITY POSTER FINE-TUNING

**HQ-Poster-100K.** To build a high-quality dataset for supervised fine-tuning, we introduce HQ-Poster-100K, a carefully filtered poster dataset. The pipeline first removes exact duplicates via MD5 hashing. To exclude posters with large credit/billing blocks, we employ an MLLM-based scoring system that asks a binary question about their presence, outputs logits, and applies Softmax to obtain probabilistic scores. Scores closer to 1 indicate better alignment with our criteria:

$$prob_x = \frac{e^{l_x}}{\sum_{x \in L} e^{l_x}}, \qquad score = \sum_{x \in L} prob_x \cdot w_x \tag{2}$$

Where set $L$ contains all the option letters, $l_x$ represents the logit for option $x$. For two-choice questions $(A, B)$, we set $w_A = 0$ and $w_B = 1$. In our pipeline, logits are computed using VLM (InternVL2.5-8B-MPO InternVL Team (2024)) with a 0.98 threshold to control filtering stringency. We then apply perceptual hashing to remove visually similar posters. The remaining posters are captioned by Gemini2.5-Flash and scored with HPS, filtering out those below 0.25 to ensure aesthetic alignment with human preferences.

To support Region-aware Calibration, HQ-Poster-100K provides precise text region masks for each poster. Since traditional OCR struggles with artistic typography, we use Gemini2.5-Flash to extract

text region coordinates. Masks are then classified as major or minor based on their relative size, denoting large and small text areas respectively, as shown in Figure2.

**Region-aware Calibration.** In poster design, harmony between text and background is crucial. Since our first stage has already improved the text rendering capability of the model, this phase shifts the focus to overall poster style. Therefore, we propose Region-aware Calibration to achieve it. Specifically, essential text carries the core message and is assigned moderate weight to ensure clarity and integration with the background; by contrast, small text—occupying minimal space and prone to rendering errors—is downweighted to prevent distracting collapse. Non-text regions, which define the visual style of the poster, receive full emphasis to guarantee a smooth transition from high-quality imagery to a unified aesthetic layout. This balanced weighting strategy allows the fine-tuned model to preserve text accuracy while strengthening the artistic integrity of the poster.

We implement this via a empirical per-pixel weight map $w(p)$:

$$
w(p) = \begin{cases} 0.6 & \text{if } p \in \text{LargeTextMask}, \\ 0.2 & \text{if } p \in \text{SmallTextMask}, \\ 1.0 & \text{otherwise}, \end{cases} \tag{3}
$$

and define our weighted loss as:

$$
\mathcal{L}_{\text{flow}}^{\text{poster}} = \mathbb{E}_{t \sim \mathcal{U}(0,1),\, x_0,\, \varepsilon} \left\| \left( v_\phi(x_t, t) - \dot{x}_t \right) \odot w \right\|_2^2, \tag{4}
$$

where "$\odot$" denotes point-wise multiplication by the weight map $w$. Different from previous scalable text rendering optimization, here we multiply the scaling factor $w$, which encourages the model to learn both crisp text information and a cohesive aesthetic style.

### 3.3 AESTHETIC-TEXT REINFORCEMENT LEARNING

**Poster-Preference-100K.** To enhance poster aesthetics and text rendering, we construct the Poster-Preference-100K dataset. Using about 20K prompts, we generate 5 images per prompt with the model after Region-aware calibration, producing 100K posters as the basis for preference pairs. With HPSv2 Wu et al. (2023), we score each group of 5 and select the highest- and lowest-scoring posters as preferred and rejected samples. Since HPSv2 evaluates only content and aesthetics, we use Gemini2.5-Flash to verify text accuracy and style consistency in the preferred posters, filtering out inconsistent pairs. This yields 6K preference pairs meeting two criteria: (i) the HPSv2 score gap ¿0.025, and (ii) complete text accuracy in preferred posters.

**Aesthetic–Text Preference Optimization.** While earlier stages ensure pixel-level text fidelity and calibrated styles, they miss higher-order trade-offs that make a poster genuinely compelling: In particular, (i) detailed preferences, such as subtle layout balance, color harmony, and typographic cohesion, which require global evaluation beyond per-pixel accuracy; (ii) even after achieving crisp text rendering, further corrective tuning is necessary to alleviate residual errors and seamlessly integrate text with the holistic aesthetic. To address these gaps, we frame poster generation as a reinforcement learning problem in this stage: the model must not only denoise accurately but also preferentially generate outputs that satisfy holistic aesthetic criteria.

Concretely, for each prompt, we sample $n$ poster variants $\{x^{(i)}\}_{i=1}^n$ under the current diffusion policy and collapse them into a single "winning" and "losing" pair via best-of-$n$ selection under the combined aesthetic–text reward $R(x)$:

$$
x^+ = \arg\max_i R\big(x^{(i)}\big), \qquad x^- = \arg\min_i R\big(x^{(i)}\big). \tag{5}
$$

We then optimize the Direct Preference Optimization (DPO) Rafailov et al. (2023) objective:

$$
\mathcal{L}_{\text{RL}}(\theta) = -\mathbb{E}_c \left[ \log \sigma \Big( \beta \Big( \log \frac{p_\theta(x^+ \mid c)}{p_{\text{ref}}(x^+ \mid c)} - \log \frac{p_\theta(x^- \mid c)}{p_{\text{ref}}(x^- \mid c)} \Big) \Big) \right], \tag{6}
$$

where $p_{\text{ref}}$ denotes the fixed reference distribution, and $p_\theta$ is learned diffusion policy parameterized by $\theta$. Because the marginal $p_\theta(x_0 \mid c)$ is intractable, we employ the ELBO over the full diffusion

Table 1: **A text quality comparison** with SOTA poster generation models, demonstrates that Poster-Craft achieves superior performance across recall, F-score, and accuracy, while only slightly below the recent closed-source Gemini2.0-Flash-Gen. We highlight the best and second metrics. Open and Close denote open-source and closed-source.

| Method | Text Recall ↑ | Text F-score ↑ | Text Accuracy ↑ |
|---|---|---|---|
| OpenCOLE Inoue et al. (2024) (Open) | 0.082 | 0.076 | 0.061 |
| Playground-v2.5 Yang et al. (2024b) (Open) | 0.157 | 0.146 | 0.132 |
| PosterMaker Gao et al. (2025) (Open) | 0.522 | 0.488 | 0.467 |
| BizGen Peng et al. (2025) (Open) | 0.689 | 0.661 | 0.641 |
| SD3.5 AI (2024) (Open) | 0.565 | 0.542 | 0.497 |
| Flux1.dev https://github.com/black-forest labs/flux (2024) (Open) | 0.723 | 0.707 | 0.667 |
| Ideogram-v2 v2. https://ideogram.ai/launch (2024) (Close) | 0.711 | 0.685 | 0.680 |
| BAGEL Deng et al. (2025) (Open) | 0.543 | 0.536 | 0.463 |
| Gemini2.0-Flash-Gen Team et al. (2023) (Close) | 0.798 | 0.786 | 0.746 |
| **PosterCraft (ours)** | 0.787 | 0.774 | 0.735 |

chain to evaluate these log-ratio rewards, following prior work Wallace et al. (2024); Wang et al. (2025). In this way, best-of-$n$ Aesthetic–Text Preference Optimization directly injects a unified preference signal into the diffusion training process.

### 3.4 VISION-LANGUAGE FEEDBACK REFINEMENT.

**Poster-Reflect-120K.** To address potential deficiencies in content and aesthetic quality of initially posters, we apply reflection optimization to improve accuracy and quality. We build the Poster-Reflect-120K dataset by generating six posters per prompt with our preference-learned model, totaling 120K images. Gemini2.5-Flash then selects the optimal poster from each set as the feedback target, required to meet three conditions: accurate prompt alignment, superior aesthetics, and correct text rendering. This process yields 5 reflection-pairs from each set of 6 generated images.

During the feedback collection phase, we gather suggestions in two key areas: ***Poster Content Suggestions*** and ***Aesthetic Style Optimization Suggestions***, with Gemini2.5-Flash analyzing both the target poster and the poster requiring optimization to provide comprehensive feedback. Additionally, to optimize our prompting strategy for both feedback and VLM fine-tuning, we implement specific guidelines: the model is instructed to perform internal comparisons without explicitly referencing the second reference poster, and feedback is structured as concrete editing instructions.

**Reflect VLM fine-tuning.** To obtain optimization feedback during inference, we construct VQA samples by embedding the original caption in the prompt alongside the generated poster requiring optimization, and using Gemini2.5-Flash-generated feedback as supervision. Specifically, this input configuration maintains consistency between training and inference phases, excluding reference target posters in both cases and using the original prompt as the baseline for suggestions. Additionally, when generating feedback with Gemini2.5-Flash, we deliberately utilize only target posters as references, omitting original captions to preserve model creativity.

**Joint Vision–Language Conditioning.** For the poster construction, iterative critique—combining visual inspection with targeted verbal feedback—is essential for refining both aesthetic content and background harmony. Inspired by this, we introduce a joint vision–language feedback loop for multimodal corrections in a unified workflow. For each generated–ground-truth pair, Gemini produces two textual reflections, $f_c$ (Poster Content Suggestions) and $f_s$ (Aesthetic Style Suggestions). Rather than appending these strings to the original prompt—which would exceed the encoder's length and degrade performance—we jointly encode them via a text encoder $E_t$, yielding $e_{c,s} = E_t(f_c, f_s)$, and then concatenate this with the original prompt embedding $e_p$ (with positional encodings to preserve order and semantics). Additionally, drawing on OmniControl Tan et al. (2024), we inject the image-level feedback signal $v_{\text{img}}$ (encoded by VAE) directly into the conditioning branch. The resulting multimodal context is:

$$c = [\, e_p;\ e_{c,s};\ v_{\text{img}} \,] \tag{7}$$

where $c$ serves as the conditioning input. Finally, we fine-tune the model under the conditional flow matching objective:

$$\mathcal{L}_{\text{flow}}^{VL}(\theta) = \mathbb{E}_{t,x_0,\varepsilon} \| v_\theta(x_t, t \mid c) - \dot{x}_t \|_2^2, \tag{8}$$

which enables the model to iteratively refine its outputs in response to structured textual reflections and semantically enriched visual feedback.

### 3.5 INFERENCE

During inference, PosterCraft generates a complete poster from a single user prompt. The prompt is first processed by a MLLM (e.g., Qwen3-based Yang et al. (2025) Magic Prompter), which enriches the input with detailed aesthetic cues to guide layout and content generation. PosterCraft then produces the poster in a unified manner, without any additional inputs or modular designs. Additionally, an VLM-based critique loop can further enhance the result by providing structured multi-modal feedback. This enables iterative refinement to improve aesthetic quality and semantic alignment.

## 4 EXPERIMENTS

### 4.1 IMPLEMENTATION

For PosterCraft, we initialize from the Flux-dev https://github.com/black-forest labs/flux (2024) backbone and train in mixed precision. Text rendering optimization involves 300K full-parameter iterations on Text-Render-2M with Adafactor Shazeer & Stern (2018) at lr = 2e-6. Stage 2 fine-tunes 6,000 steps on HQ-Poster-100K using Adafactor (lr = 1e-5) with per-pixel flow-matching weights 0.6, 0.2, 1.0. For reinforcement learning, we sample n=5 candidates per prompt and optimize via AdamW Loshchilov & Hutter (2017) (lr = 1e-4) for 1500 steps, tuning only LoRA (rank 64). In vision-language feedback refinement, dual-language reflections $f_c$ and $f_s$ are encoded by T5 Raffel et al. (2020), followed by LoRA (rank 128) fine-tuning under conditional flow-matching loss for 6000 steps using AdamW (lr = 1e-4). We employ Internvl3-8B InternVL Team (2025) for feedback generation, fine-tuning 2 epochs and using temperature 0 at inference. More details about the dataset can be found in the supplementary material.

### 4.2 QUANTITATIVE RESULTS AND COMPARISONS

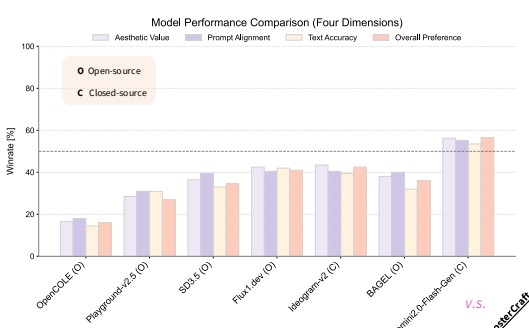

Figure 5: **User study comparisons** between PosterCraft and both SOTA open-source and closed-source models. It consistently outperforms all open-source baselines and several proprietary systems cross multiple dimensions, with performance marginally below that of the leading closed-source model, Gemini2.0-Flash-Gen.

We conduct a quantitative comparison of PosterCraft against seven leading models, both open-source and commercial. Using Gemini2.0-Flash-Gen Team et al. (2023), we randomly generate 100 aesthetic poster prompts—balanced across short, medium, and long lengths—and sample three outputs per model, yielding 300 test images. Posters are generated with OpenCOLE, Playground-v2.5, SD3.5, Flux1.dev, Ideogram-v2, BAGEL, and Gemini2.0-Flash-Gen. We then apply the OCR engine of the SOTA VLM to each image and report three precision-oriented metrics—text recall, text F-score, and text accuracy—averaged over all 300 samples.

As shown in Tab. 1, PosterCraft achieves substantially higher recall and F1 scores, capturing a more complete character set, while its accuracy surpasses leading open-source methods (Flux1.dev, SD3.5) and competitive commercial solutions (Ideogram-v2). The table also includes poster-specific generators BizGen and PosterMaker, for which we provide the same five representative layouts (and no product image for PosterMaker); despite their more structured inputs, PosterCraft still attains better recall, F-score, and accuracy. PosterCraft's performance closely approaches that of some mature SOTA commercial systems such as Gemini2.0-Flash-Gen, demonstrating that its text-rendering not only advances academic benchmarks but also offers near production-level competitiveness. Besides, inspired by previous works Chen et al. (2024a); Esser et al. (2024), we conduct the user study to assess model quality. Twenty experienced poster designers evaluate outputs across multiple dimensions, as shown in Fig. 5. Experiments show that our method significantly enhances aesthetics and text rendering over the base model Flux1.dev, and outperforms all state-of-the-art open-source and several closed-

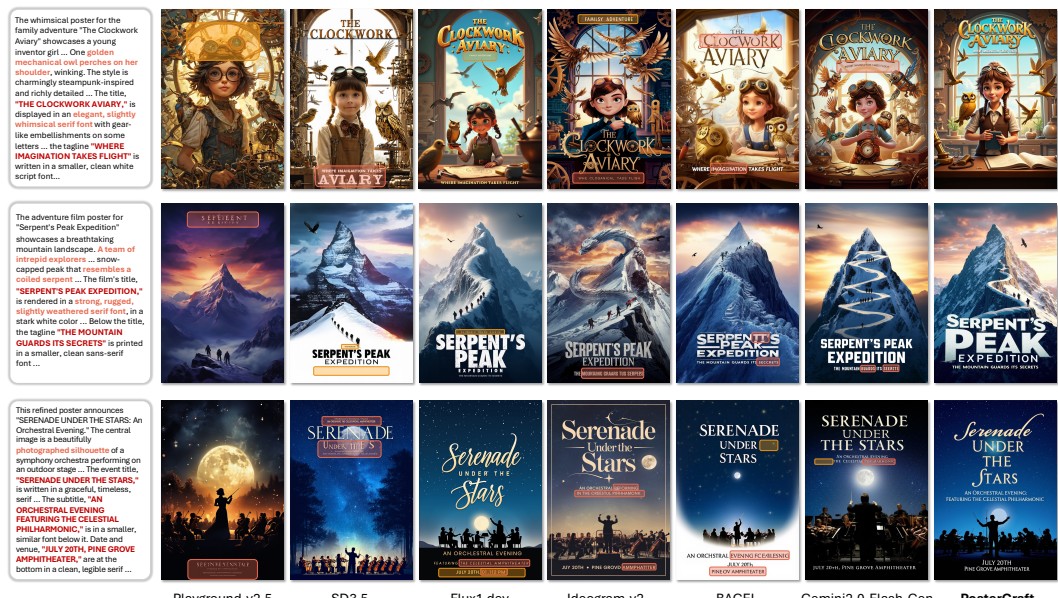

Figure 6: **Visual comparison** of different model outputs. Red boxes highlight misspelled or distorted text, while yellow boxes indicate redundant or missing text elements. Within the prompts, orange text denotes content and style requirements, and red text indicates textual elements. From the visual comparison, it is evident that our method achieves superior aesthetic appeal compared to existing state-of-the-art approaches. In terms of text presentation, our rendered fonts blend more naturally with the scene content, and text rendering errors are nearly eliminated.

source models across all evaluation metrics. Additionally, PosterCraft performs only slightly below Gemini2.0-Flash-Gen, validating the effectiveness of our unified workflow in fully unlocking the generative potential of the powerful baseline model. More detailed evaluation strategies and experiments on other benchmarks can be found in the supplementary material.

## 4.3 QUALITATIVE RESULTS AND COMPARISONS

To further evaluate the superiority of vision, we conduct a comprehensive visual comparison between PosterCraft and six other models—four open-source and two commercial systems, as shown in Fig.6. Playground-v2.5 completely fails in text rendering, while SD3.5 renders titles only partially, with both showing poor aesthetics and weak prompt adherence (e.g., animated styles). Flux1.dev and BAGEL improve title rendering but still contain textual errors and fall short in aesthetics and prompt alignment, missing details such as "One golden mechanical owl perches on her shoulder." Among proprietary models, Ideogram-v2 and Gemini2.0-Flash-Gen show only minor text errors in small text and achieve superior aesthetics, but both have prompt adherence issues: Ideogram-v2 misrepresents mountain formations, Gemini lacks photorealism, and neither generates silhouette effects in the third image. More comparisons appear in the supplementary material.

## 5 ABLATION STUDY

In this section, we validate our workflow by independently assessing each stage, keeping all parameters and conditions identical to the preceding experiments. **Text Rendering Optimization is critical for both accuracy and**

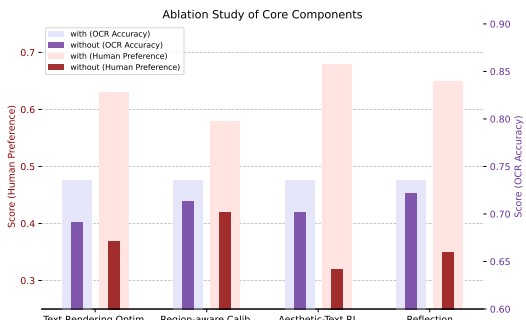

Figure 7: **Abl. experiments on the core components of our workflow**: Removing any of these four improvements results in sustained declines in OCR accuracy (purple) and human preference (peach), showing our design and optimizations are effective and our motivation valid.

**perception.** Text Rendering Optimization ensures clarity and fidelity. Its removal leads to drops in both OCR accuracy and human preference, confirming that large-scale, high-quality text data significantly improves text rendering. The diverse and realistic backgrounds also preserve visual quality, while models without this optimization often fail to maintain legibility and accuracy, as shown in Fig.4 and Fig. 7. **Region-aware Calibration improves poster consistency.** Fig. 7 demonstrates that it helps the model adapt to spatial context and balance text-background. Without it, all regions are treated equally, weakening stylistic coherence in visually complex posters while suffering text bias. **Aesthetic-Text Reinforcement Learning and Reflection refine outputs with vision-language feedback.** As shown in Fig. 7, Aesthetic-Text RL directly enhances the harmony between aesthetic appeal and textual accuracy, while Reflection further improves overall quality through iterative vision-language guidance. We provide more detailed ablation studies in the supplementary material.

## 6 CONCLUSION

PosterCraft presents a unified, cascaded workflow that integrates scalable text-rendering optimization, poster fine-tuning, RL-driven aesthetic enhancement, and joint vision–language feedback, demonstrating powerful foundation model can directly produce posters with both striking visuals and precise text. Our fully automated dataset pipelines support scalable, task-specific training, and we achieve substantial gains over open-source baselines, approaching leading commercial quality.

## 7 ETHICS STATEMENT

This work does not involve human subjects, personally identifiable information, or sensitive private data. All datasets used in this study were either automatically constructed through synthetic pipelines (e.g., Text-Render-2M, Poster-Preference-100K, Poster-Reflect-120K) or collected from publicly available poster resources (HQ-Poster-100K) with automated filtering to ensure quality and compliance.

The HQ-Poster-100K dataset contains materials protected by third-party copyrights. Their inclusion follows the principle of fair use under copyright law and is intended solely for non-commercial scientific research. We do not claim copyright ownership of these materials, and users are responsible for ensuring that any use complies with applicable laws and regulations. The publishers of this project assume no responsibility for copyright disputes arising from use of the dataset.

Our method focuses on aesthetic poster generation and does not target manipulative, deceptive, or harmful applications. We are committed to releasing our datasets and models under appropriate open-source licenses to support transparency, reproducibility, and fair access for the research community.

## 8 REPRODUCIBILITY STATEMENT

We have made substantial efforts to ensure the reproducibility of our work. The main paper and supplementary materials provide detailed descriptions of the proposed framework, dataset construction pipelines, training configurations, and evaluation protocols. Hyperparameters and ablation studies are explicitly documented to facilitate independent verification. With the growing availability of alternative models, many of our data filtering designs based on closed-source models can be easily reproduced using state-of-the-art open-source vision–language models. In addition, we include the data processing scripts, evaluation code, and all prompts used in testing in the supplementary materials to further support transparency and reproducibility. All datasets and models used in our experiments are described with sufficient detail to ensure full reproducibility and to enable future research extensions.

**Acknowledgments.** This work is supported by the Guangdong Science and Technology Department (No. 2024ZDZX2004), the National Natural Science Foundation of China (Project No.82572383), and Adobe Research Gifted Grant.

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

## A  APPENDIX

This is supplementary material for *PosterCraft: Rethinking High-Quality Aesthetic Poster Generation in a Unified Framework.*

We present the following materials in this supplementary material:

- Sec.A.1 Detailed dataset construction on Text-Render-2M.

- Sec.A.2 More information about the automatic processing pipeline for HQ-Poster-100K.

- Sec.A.3 More examples and explanations of the Poster-Preference-100K.

- Sec.A.4 Detailed illustration of the Poster-Reflect-120K.

- Sec.A.5 Gemini for OCR calculation and preference evaluator.

- Sec.A.6 Discussion about reflection and instruction-based editing.

- Sec.A.7 An extra evaluation strategy comparing PosterCraft with other methods.

- Sec.A.8 More performance comparisons of PosterCraft and other methods on different task benchmarks.

- Sec.A.9 More key ablation experiments and effectiveness demonstrations of our framework.

- Sec.A.10 Additional visual results and comparisons generated by our PosterCraft.

- Sec.A.11 Limitations.

- Sec.A.12 Future work.

- Sec.A.13 Usage of LLM.

- Sec.A.14 Generalizability beyond Flux-dev: Experiments on SD3.5.

- Sec.A.15 Why a unified PosterCraft framework works.

- Sec.A.16 Dataset availability and statistical analysis of our main datasets.

## A.1 TEXT-RENDER-2M CONSTRUCTION PIPELINE

In this section, we provide a detailed overview of the automated construction process behind Text-Render-2M, a large-scale synthetic dataset designed to improve text rendering quality in the baseline model. This dataset plays a critical role in the text rendering optimization stage of our workflow. By overlaying diverse textual elements onto high-resolution background images, it allows the model to learn accurate text generation while preserving its ability to represent rich visual content.

Multi-Instance Text Rendering Each image contains a variable number of independently placed text instances, typically ranging from one to three. This multi-instance setup introduces compositional complexity and better reflects natural layouts found in aesthetic posters.

Text Content Generation Text content is synthesized using a hybrid strategy:

- A majority of the samples are generated using template-based grammars, with phrases constructed from predefined structures filled with curated vocabulary lists.
- A smaller portion uses random alphanumeric strings to simulate noisy or unstructured textual inputs.

The generator supports rich variations in punctuation, casing , and structure (e.g., both single-word and short-phrase constructions), ensuring linguistic diversity.

Font Selection and Style Variability Fonts are randomly drawn from a categorized library containing both standard and artistic typefaces. When multiple styles are available, a roughly even split is enforced between classic and stylized fonts. The system filters fonts that do not support lowercase letters to avoid invalid renderings. This selection mechanism ensures visual variability while maintaining text legibility.

Layout and Placement Strategy Text is positioned using a 3×3 grid-based partitioning scheme (e.g., top-left, center, bottom-right). Before final placement, bounding box collision checks are performed. If an overlap is detected, the system will retry placement with different positions or reduced font

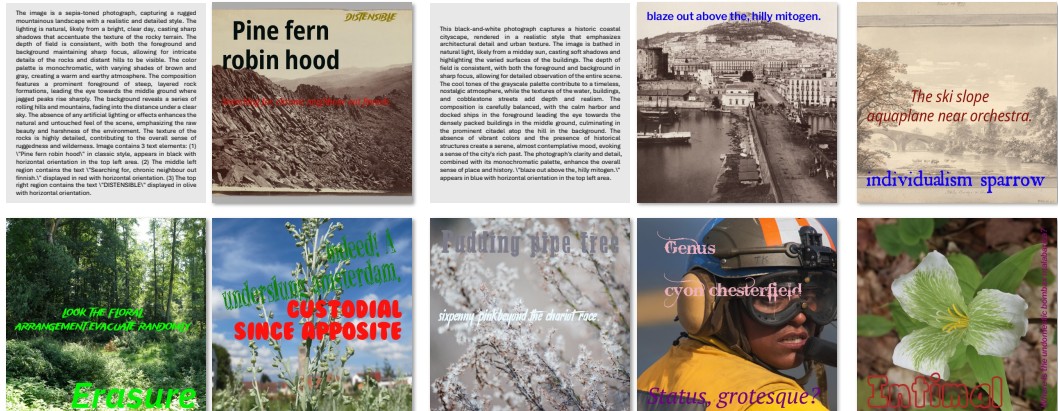

Figure 8: More samples are shown, which are high-quality paired samples in Text-Render-2M.

sizes, often within 3–5 attempts per instance. This strategy enables dense yet legible arrangements while minimizing visual clutter.

Orientation and Alignment Orientation options include:

- Horizontal,
- Vertically rotated (rotated 90 degrees),
- Vertically stacked (one character per line).

In the horizontal mode, a small proportion of texts receive a random rotation, and a subset of longer text fragments are automatically wrapped across multiple lines. Alignment is randomly selected among left, center, or right justification.

Prompt Generation Each image is paired with a structured natural language prompt. Prompts include:

- The text content,
- Position (e.g., "bottom right"),
- Orientation (e.g., "vertically stacked"),
- Color category, and
- Optionally (included in 50% of cases), the font style.

When multiple texts are rendered, their prompts are numbered and concatenated. If no text is successfully rendered, a fallback prompt indicating the absence of text is generated. We provide a number of samples to view in Fig.8.

## A.2 AUTOMATIC PROCESSING PIPELINE FOR HQ-POSTER-100K

For collecting high-quality poster images, we design an automated image filtering and annotation pipeline for HQ-Poster-100K utilizing three MLLMs:

(1) MLLM Scorer: We employ Internvl2.5-8B-MPO with a binary classification task to filter out posters containing extensive Credit Blocks, Billing Blocks, or "4K ultrahd" cover texts. The complete prompt is shown in Fig.17.

(2) Gemini Caption Generation: We utilize Gemini2.5-Flash-Preview-04-17 for automated poster caption annotation, systematically describing the poster content, visual style, and textual elements. The complete prompt structure is illustrated in Fig.18

(3) Gemini Mask Generation: The final step in poster data collection involves generating text region masks. We employ Gemini2.5-Flash-Preview-04-17's robust OCR capabilities to extract text masks

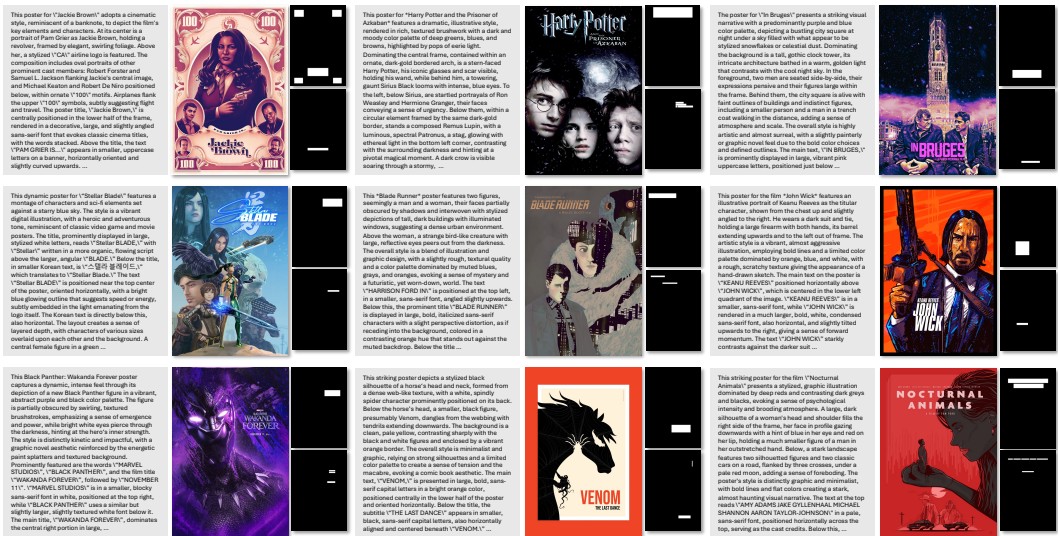

Figure 9: More pictures are shown, which are high-quality paired samples with masks in HQ-Poster-100K.

from posters, which are subsequently used for Region-aware Calibration. The prompt is shown in Fig.19

In addition, we provide more masks, images, prompts triplets in Fig.9 to demonstrate the advantages of our dataset.

### A.3 EXPLANATIONS OF THE POSTER-PREFERENCE-100K

In constructing the Poster-Preference-100K dataset, we first utilize HPSv2 to filter out poster pairs that meet aesthetic requirements and exhibit sufficient diversity. Subsequently, we employ Gemini2.5-Flash-Preview-04-17 to evaluate the alignment between the Best of 5 posters and their corresponding prompts. The complete prompt is shown in Fig.20 Our criteria require that text in the Best of 5 posters must be completely accurate while maximizing alignment with both the content and aesthetic style requirements specified in the prompt. We implement a binary classification system where the model assigns 0 to unqualified samples and 1 to samples meeting all requirements. Fig.10 presents additional preference pairs, demonstrating that our pipeline built upon HPSv2 and Gemini effectively constructs preference data.

### A.4 ILLUSTRATION OF THE POSTER-REFLECT-120K

In constructing the Poster-Reflect-120K dataset, we first employ a preference-optimized model to generate 6 posters per prompt, totaling 120K posters. Subsequently, our feedback collection pipeline utilizes Gemini2.5-Flash-Preview-04-17 in two phases: first to select the Best of 6 posters as optimization targets, and second to gather direct feedback.

(1) Best of 6 Selection: We sequentially input six images to Gemini, which selects the optimal image based on predetermined priorities, returning the index number of the best image. The complete prompt is shown in Fig.21.

(2) Feedback Collection: We input two images sequentially - the first being the image requiring improvement, and the second being the optimization target selected in phase (1). Feedback is collected across two dimensions: "Poster Content Suggestions" and "Aesthetic Style Optimization Suggestions". Each image pair receives 10 pieces of feedback through 5 Gemini requests. The complete prompt is shown in Fig.22.

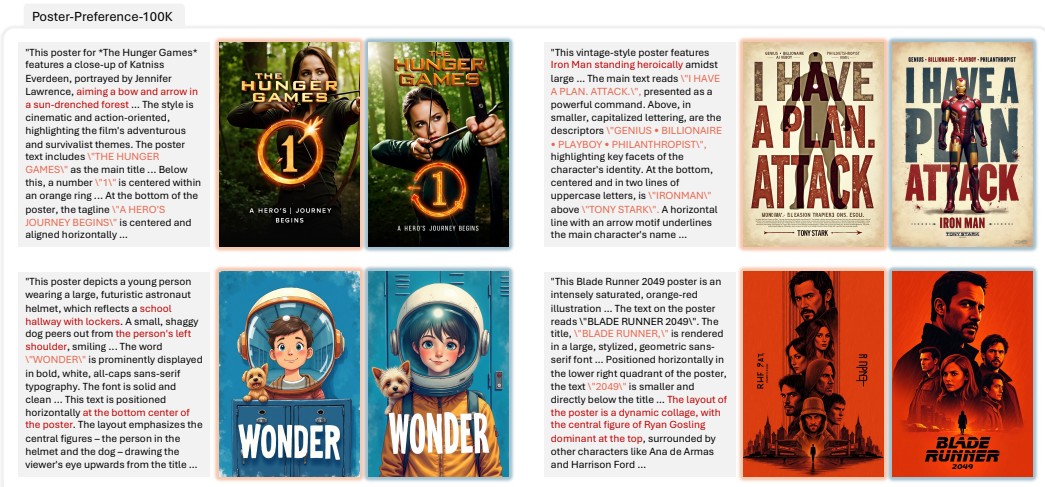

Figure 10: **Additional Preference Pairs** in Poster-Preference-100K. The images on the left are Rejected Samples, while those on the right are Preferred Samples. Orange text indicates textual content, and red text corresponds to content, style, or layout requirements.

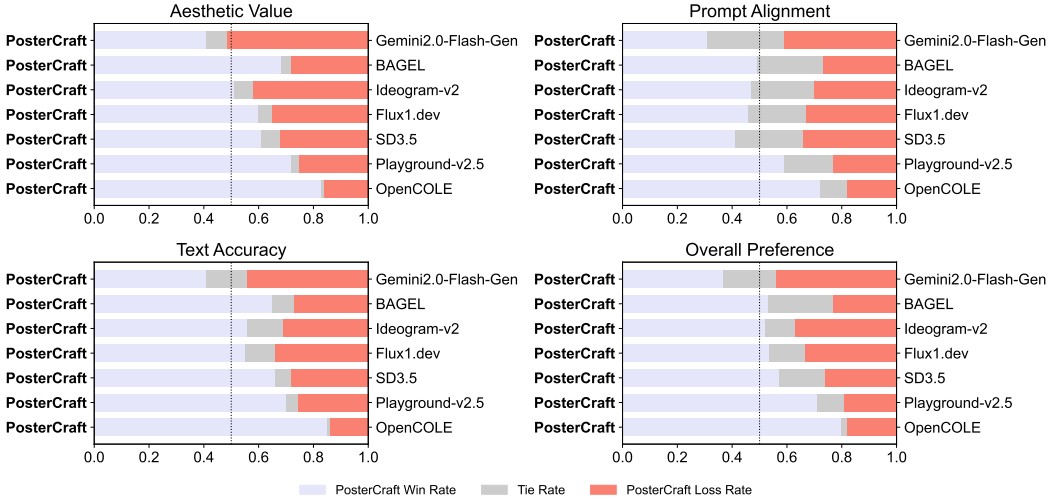

Figure 11: **Gemini serves as an authoritative evaluator for human preference comparisons** across our method and other baselines. PosterCraft outperforms most state-of-the-art generative models in aesthetic coherence, prompt alignment, text rendering, and overall preference. It achieves performance nearly on par with the leading commercial model Gemini2.0-Flash-Gen in text rendering, while showing only a slight gap in other aspects.

## A.5 GEMINI FOR OCR CALCULATION AND PREFERENCE EVALUATOR

During the experimental phase, we employ Gemini2.5-Flash-Preview-05-20 as both an OCR metric calculator and a multi-dimensional preference evaluator. The robust perceptual and comprehension capabilities of Gemini2.5-Flash establish a solid foundation for our experimental accuracy.

(1) OCR Evaluator: Despite the significant challenges that artistic fonts in posters pose to traditional OCR algorithms, Gemini2.5-Flash-Preview-05-20, as a state-of-the-art MLLM model, accurately extracts textual information from images. We further utilize Gemini2.5-Flash's reasoning and mathematical capabilities to directly compute and output OCR metrics. The specific prompt is shown in Fig.23.

(2) Multi-Dimensional Preference Evaluator: Gemini2.5-Flash demonstrates superior assessment capabilities with its reasoning abilities. We present two images side by side and instruct Gemini2.5-

Table 2: **Comparison on the our proposed dataset Geng et al. (2025) using HPSv3 Ma et al. (2025).** We highlight the best and second performance. The upward arrow (↑) indicates that higher values are better.

| Method | HPSv3 ↑ |
|---|---|
| OpenCOLE Inoue et al. (2024) (Open) | 8.2280 |
| PosterMaker Gao et al. (2025) (Open) | 8.5825 |
| Playground-v2.5 Yang et al. (2024b) (Open) | 8.7831 |
| BizGen Peng et al. (2025) (Open) | 9.4879 |
| SD3.5 AI (2024) (Open) | 10.3104 |
| Flux1.dev https://github.com/black-forest labs/flux (2024) (Open) | 10.4066 |
| Ideogram-v2 v2. https://ideogram.ai/launch (2024) (Close) | 10.0911 |
| HiDream-I1-Full Cai et al. (2025) (Open) | 10.4533 |
| BAGEL Deng et al. (2025) (Open) | 8.9116 |
| Gemini2.0-Flash-Gen Team et al. (2023) (Close) | 10.5251 |
| **PosterCraft (ours)** | 10.7402 |

Flash to select from "L", "R", or "None", representing left image superior, right image superior, or indeterminate (either both excellent or both inadequate). The specific prompt is shown in Fig.24 and Fig.25.

## A.6 DISCUSSION ON THE DIFFERENCES BETWEEN REFLECTION AND INSTRUCTION-BASED EDITING

While our refinement stage may appear similar to instruction-based editing, the underlying mechanisms and objectives differ substantially. We summarize the key distinctions as follows:

- **Source of instruction.** In instruction-based editing, the model typically receives explicit, human-written commands describing object-level changes (e.g., "remove the chair" or "change background color"). By contrast, our reflection mechanism relies on an offline aesthetic model that generates structured feedback from imperfect images. This enables multi-dimensional improvement, covering aesthetics, typography, and layout balance, rather than focusing solely on localized object edits or simple manipulations.

- **Role of the original prompt.** Instruction-based editing often discards or overrides the initial text-to-image (T2I) prompt. In our design, the original T2I prompt remains central, while the reflection-derived instruction serves only as auxiliary guidance. This ensures that semantic alignment with the initial user intent is preserved, while aesthetics and overall visual quality are enhanced.

- **Objective difference.** Instruction-based editing aims for literal compliance with user-specified edits. In contrast, reflection targets holistic enhancement—refining text rendering, composition, and aesthetic appeal. In this sense, reflection is closer to an iterative self-improvement loop rather than one-shot command execution.

## A.7 MORE EVALUATION STRATEGY

Inspired by PixArt-Σ Chen et al. (2024a), we further employ Gemini2.0-Flash as an automated evaluator to provide strict preference comparisons across models, as illustrated in Fig. 11. Leveraging carefully designed evaluation prompts, Gemini consistently confirms that PosterCraft achieves superior aesthetic quality, faithful prompt alignment, and accurate text rendering compared to both open-source and several closed-source baselines. Although slightly behind the commercial Gemini2.0-Flash-Gen in text rendering, PosterCraft remains competitive overall, demonstrating the robustness of our unified workflow.

To further strengthen the aesthetic and prompt alignment preference evaluation, we additionally benchmarked PosterCraft against existing methods using HPSv3 on the prompts from our proposed poster generation benchmark (Tab. 2). In this comparison we also include poster-oriented systems such as BizGen and PosterMaker, adapting them to our text-only setting by supplying five representative poster layouts and, for PosterMaker, disabling the product-image input. PosterCraft achieves the highest HPSv3 score (10.7402), surpassing both open-source and commercial baselines, with

Table 3: **Quantitative results on CVTG-2K Du et al. (2025) dataset.** We highlight the `best` and `second` performance for each metric. The upward arrow (↑) indicates that higher values are better.

| Model | Word Accuracy ↑ | NED ↑ | CLIPScore ↑ | VQAScore ↑ |
|---|---|---|---|---|
| FLUX.1-dev https://github.com/black-forest labs/flux (2024) | 0.4965 | 0.6879 | 0.7401 | 0.8886 |
| AnyText Tuo et al. (2023) | 0.1804 | 0.4675 | 0.7432 | 0.6935 |
| TextDiffuser-2 Chen et al. (2023b) | 0.2326 | 0.4353 | 0.6765 | 0.5627 |
| RAG-Diffusion Chen et al. (2024c) | 0.2648 | 0.4498 | 0.6688 | 0.6397 |
| 3DIS Zhou et al. (2024b) | 0.3813 | 0.6505 | 0.7767 | 0.8684 |
| TextCrafter Du et al. (2025) | 0.7370 | 0.8679 | 0.7868 | 0.9140 |
| **PosterCraft (Ours)** | 0.5814 | 0.7581 | 0.7493 | 0.9007 |
| **PosterCraft (Ours) with TextCrafter** | 0.7544 | 0.8863 | 0.7952 | 0.9241 |

Table 4: **Comparison on the LongText-Bench (English) dataset Geng et al. (2025).** We highlight the `best` and `second` performance. The upward arrow (↑) indicates that higher values are better.

| Method | LongText-Bench-Eng ↑ |
|---|---|
| Janus-Pro Chen et al. (2025d) | 0.019 |
| BLIP3-o Chen et al. (2025b) | 0.021 |
| Kolors 2.0 Team (2024b) | 0.258 |
| PosterMaker (without image input) Gao et al. (2025) | 0.315 |
| BAGEL Deng et al. (2025) | 0.373 |
| BizGen (random layouts input) Peng et al. (2025) | 0.478 |
| HiDream-I1-Full Cai et al. (2025) | 0.543 |
| OmniGen2 Wu et al. (2025) | 0.561 |
| FLUX.1-dev https://github.com/black-forest labs/flux (2024) | 0.607 |
| X-Omni Geng et al. (2025) | 0.900 |
| **PosterCraft (Ours)** | 0.631 |

Gemini2.0-Flash-Gen ranking second (10.5251). That PosterCraft outperforms these task-specific poster generators while using only a single text prompt further emphasizes the strength of our unified backbone design. These results highlight the superiority of our unified workflow in capturing human-preferred aesthetics beyond text accuracy and layout coherence. Taken together, our evaluation covers a comprehensive spectrum—from user studies, to win-rate analysis with state-of-the-art VLM evaluators, to objective metrics of poster text accuracy and aesthetic quality—demonstrating that PosterCraft delivers outstanding performance in aesthetic poster generation across multiple perspectives.

## A.8 MORE BENCHMARK COMPARISONS

Our primary task is poster generation, and in the main manuscript we have already demonstrated that PosterCraft can match closed-source models in that domain. Here we additionally report results on more general text-rendering benchmarks to show that our training workflow also brings consistent gains beyond aesthetic posters. On the partitioned text dataset CVTG-2K Du et al. (2025), PosterCraft achieves Word Accuracy = 0.5814, NED = 0.7581, CLIPScore = 0.7493, and VQAScore = 0.9007. Although we do not outperform methods specifically designed for text rendering (for instance, TextCrafter, which uses attention- and partition-aware strategies), PosterCraft surpasses almost all open-source baselines (such as FLUX.1-dev, AnyText, TextDiffuser-2, RAG-Diffusion, and 3DIS). Moreover, when we apply the training-free TextCrafter optimizer on top of PosterCraft, the combined system (PosterCraft + TextCrafter) further improves all four CVTG-2K metrics and clearly surpasses TextCrafter itself, as shown in Table 3. This confirms that our four-stage training pipeline has already unlocked strong text-rendering capacity in the backbone and remains fully compatible with specialized test-time plug-ins when desired. Notably, we also significantly outperform our base model FLUX, which demonstrates that our proposed training workflow yields substantive improvements even on tasks different from poster generation. This strongly indicates that our workflow possesses robust generalization and resilience.

To further validate PosterCraft's capability in longer text generation in image contexts, we evaluate it on the LongText-Bench (English) dataset. PosterCraft attains a score of 0.631, ranking second among public models (only X-Omni is ahead with a score of 0.900). Although we do not surpass

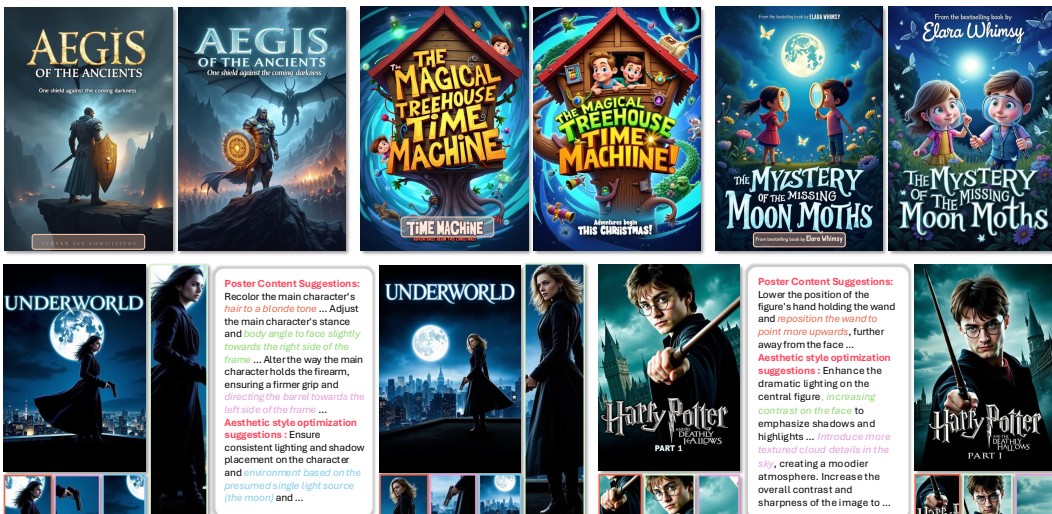

Figure 12: **Aesthetic-text reinforcement learning (top row) and vision-language feedback (bottom row) qualitative comparisons.** orange boxes denote the text biases. Different color fonts represent key feedback information from VLM. The top examples demonstrate that our reinforcement learning stage effectively improves overall aesthetic quality and partially corrects text rendering errors after supervised fine-tuning. The bottom examples illustrate the impact of vision-language reflection, where feedback from a VLM is integrated into the generation loop. This results in noticeable enhancements to both visual aesthetics and semantic coherence in the final poster outputs.

X-Omni, we outperform many prior methods (such as Janus-Pro, BLIP3-o, Kolors 2.0, BAGEL, HiDream-I1-Full, OmniGen2, FLUX.1-dev, etc.), including the very strong HiDream-I1-Full. For poster-specific baselines BizGen and PosterMaker (needing layout), we randomly sample one of five predefined layouts per prompt and omit the product image for PosterMaker—so that they match the testing form of LongText-Bench. Given that X-Omni is built on a much larger backbone and substantially more text-centric data, this gap is expected and mainly reflects differences in base-model capacity and supervision rather than a limitation of PosterCraft. Yet the fact that we outperform most open models, and significantly surpass our base model, highlights the generalizability and effectiveness of our method. These additional benchmark results serve as supportive evidence, reinforcing the argument that PosterCraft is not limited to poster generation, but also has promise across broader visual text generation tasks.

## A.9 MORE ABLATION EXPERIMENTS AND EFFECTIVENESS DEMONSTRATIONS

### A.9.1 AESTHETIC-TEXT REINFORCEMENT LEARNING AND VISION-LANGUAGE REFLECTION

**Aesthetic-Text Reinforcement Learning.** The reinforcement learning stage significantly boosts visual appeal and human preference. By optimizing higher-order aesthetic cues (e.g., layout balance, color harmony, typographic cohesion), it enables the model to generate posters that are more visually compelling and coherent. The top-row results in Fig. 12 clearly show these improvements: reinforcement learning corrects text rendering errors left after supervised fine-tuning and enhances holistic preference, confirming its effectiveness.

**Vision-Language Reflection.** The reflection stage integrates structured feedback from a vision-language model (VLM) into the generation loop. This mechanism allows the system to iteratively refine stylistic integration, semantic alignment, and overall aesthetic appeal. The bottom-row examples in Fig. 12 highlight that reflection leads to more consistent layouts and improved text-background harmony, particularly in visually complex posters.

These ablations and Fig.7 in the manuscript demonstrate that both stages—reinforcement learning and vision-language reflection—are indispensable for unlocking the full potential of foundation diffusion models in end-to-end aesthetic poster generation.

| Stage-4 (Reflection) | Feedback generator | Feedback Quality (BERTScore-F1 ↑) | Win rate ↑ |
|---|---|---|---|
| *No (w/o Stage-4)* | – | N/A | 0.33 |
| Yes | InternVL3 (original) | $0.8641 \pm 0.0099$ | 0.30 |
| Yes | InternVL3 (fine-tuned) | $0.8957 \pm 0.0095$ | 0.37 |

Table 5: Ablation on the VLM used to produce Stage-4 feedback. Fine-tuning the VLM with high-quality, domain-specific feedback pairs improves both the quality of feedback text and downstream human preference.

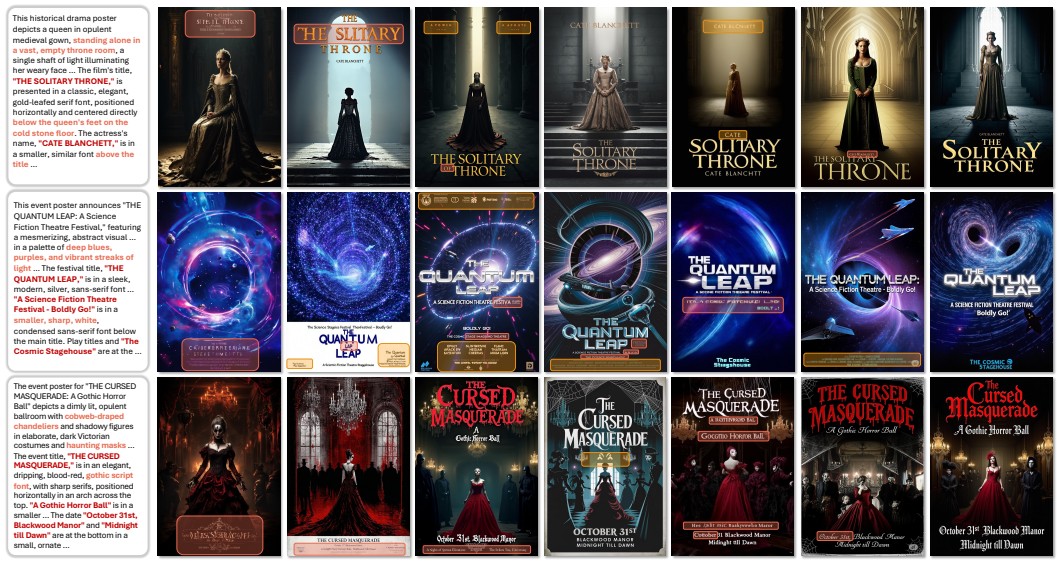

Figure 13: **Visual comparison** of different model outputs. Red boxes highlight misspelled or distorted text, while yellow boxes indicate redundant or missing text elements. Within the prompts, orange text denotes content and style requirements, and red text indicates textual elements. It indicates that our method significantly outperforms existing SOTA approaches in generating high-quality posters under long-prompt conditions, with notably improved prompt alignment. In terms of text rendering, our model produces fonts that align closely with the visual context of the scene, with minimal rendering errors.

### A.9.2 THE NECESSITY OF FEEDBACK MODEL FINE-TUNING

Fine-tuning the VLM used for Stage-4 feedback substantially improves both the feedback text and downstream generation quality. Compared with the original VLM, our fine-tuned version yields higher feedback quality (BERTScore-F1 $0.8957 \pm 0.0095$ vs. $0.8641 \pm 0.0099$, +0.0316 absolute, $\sim$3.66% relative). A higher BERTScore-F1 indicates better semantic alignment between our generated feedback and reference (high-quality) feedback: it means that the fine-tuned VLM's feedback is more semantically faithful to the gold feedback, and it increases the human win rate of generated posters from 0.30 to 0.37 (+0.07 absolute, $\sim$23.33% relative). Notably, using the original VLM hurts Stage-4 compared with removing reflection entirely (0.30 vs. 0.33), indicating that non-domain or ill-formatted feedback can misguide the reflection loop. Enforcing a structured schema and training the VLM on high-quality, domain-specific feedback pairs is therefore essential for stable, effective reflection.

### A.10 ADDITIONAL VISUAL COMPARISON AND EXAMPLES

In the long-prompt setting of Fig.13, PosterCraft demonstrates clear advantages in both prompt alignment and aesthetic quality. Our model consistently integrates complex scene descriptions with textual elements, producing posters that closely follow the narrative and stylistic cues in the prompts. Competing models often struggle with either missing or distorted text (e.g., Playground-v2.5, SD3.5, Flux1.dev, Gemini2.0-Flash-Gen), or fail to faithfully capture scene details (e.g.,

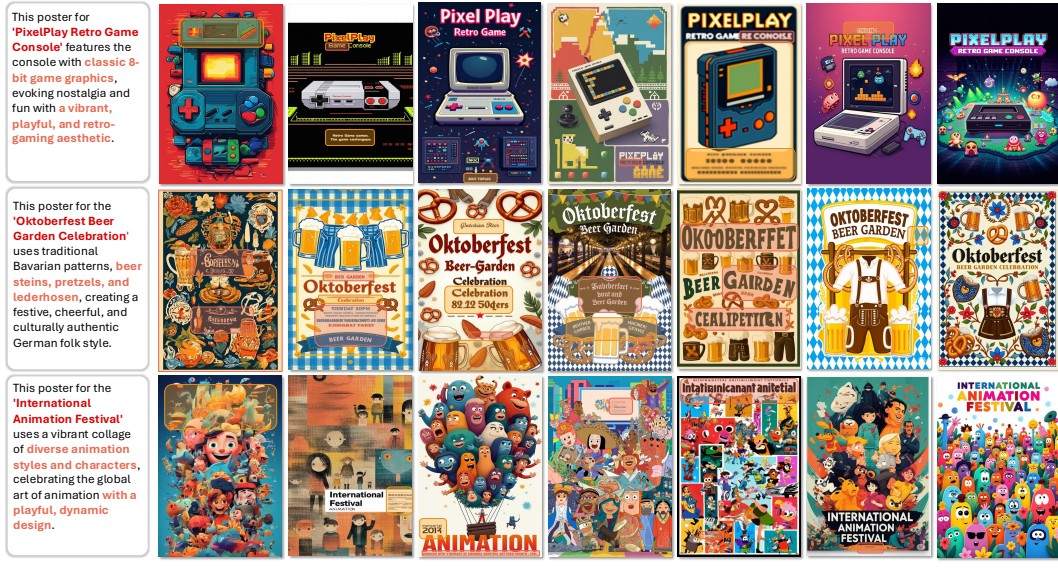

Figure 14: **Visual comparison** of different model outputs. Red boxes highlight misspelled or distorted text, while yellow boxes indicate redundant or missing text elements. Within the prompts, orange text denotes content and style requirements, and red text indicates textual elements. Compared to other methods, our approach produces cleaner layouts, better theme alignment, and more accurate text rendering under short prompts.

BAGEL, Ideogram-v2). By contrast, PosterCraft generates outputs with coherent layouts, accurate text rendering, and strong adherence to detailed style requirements, achieving more natural visual-text alignment even under demanding long-prompt conditions.

In the short-prompt setting of Fig.14, PosterCraft maintains a strong balance between visual appeal and accurate text rendering. It consistently integrates title and scene elements, for example, seamlessly embedding 'PixelPlay Retro Game Console' or 'International Animation Festival' in stylized compositions. Competing models often display legibility issues (e.g., SD3.5, Flux1.dev), omit key poster elements (e.g., BAGEL, Ideogram-v2), or suffer from text-scene disconnection (e.g., Playground-v2.5). While Gemini2.0-Flash-Gen performs well in text rendering, it still suffers from aesthetic limitations, often producing visually monotonous outputs with missing or underdeveloped design elements. Our results stand out with their vibrant layout, theme adherence, and natural visual-text coherence, even under minimal input conditions.

We provide additional visual results generated by our PosterCraft framework to further demonstrate its capability in producing high-quality, aesthetically consistent posters. As shown in Fig.15 and Fig.16, our model is able to seamlessly integrate text and imagery without requiring any external layout templates or modular refinement. The generated posters exhibit strong visual coherence—text elements are not only stylistically aligned with the visual content but are sometimes cleverly embedded into the composition, enhancing the overall design fluency. From an aesthetic perspective, the model captures genre-specific styles across diverse themes, such as cinematic sci-fi, educational charts, cultural festivals, and commercial advertisements. It achieves a fine balance between visual richness and layout readability, effectively modeling principles such as symmetry, emphasis, and hierarchy. These results underscore PosterCraft's potential as a powerful end-to-end tool for automatic poster generation with minimal input while maintaining high visual and semantic fidelity.

## A.11    LIMITATIONS

In this work, we propose PosterCraft to explore how unified workflow design can unlock the aesthetic design potential of powerful foundation models. Our results demonstrate that with carefully crafted design strategies, the model's capabilities can be significantly enhanced—making it competitive with leading proprietary commercial systems. This validates the soundness of our motivation. However, our approach is not without limitations. Specifically, our model is fundamentally built

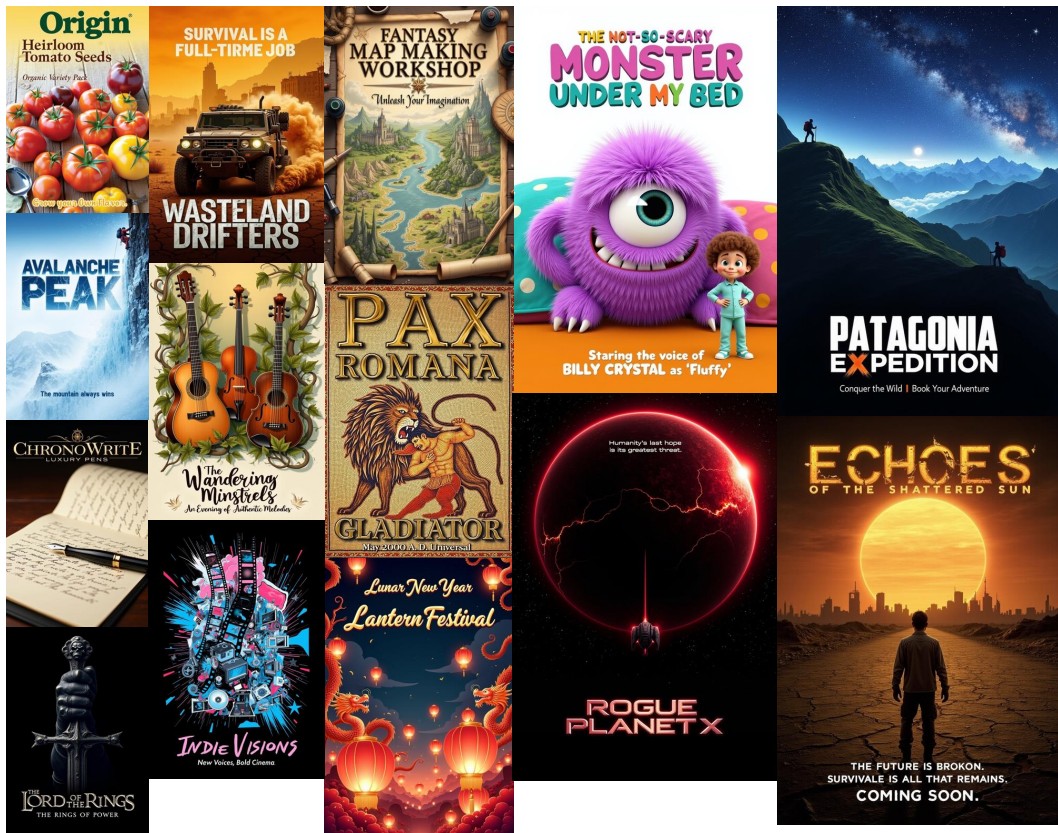

Figure 15: **Examples generated by our PosterCraft** demonstrating high diversity and aesthetic quality across themes including education, entertainment, and science fiction. All generation results showcase genre-specific fidelity, text rendering, and layout aesthetic.

upon the current flux1.dev baseline. As such, if the pre-trained flux model has never encountered certain types of samples or contains significant flaws, our method may not be able to fully correct these shortcomings. That said, our workflow is highly unified and readily transferable to stronger baselines, ensuring full compatibility with other models in the community.

### A.12 FUTURE WORK

In future work, we plan to enhance our unified workflow in several directions. First, we will explore integrating PosterCraft with stronger backbone models and broader pretraining data, so that our four-stage pipeline can further close the gap to the most advanced commercial systems while remaining architecture-agnostic. Second, we aim to scale up training with larger and more diverse datasets, and to adapt our text–aesthetic preference design to product-poster scenarios where both product images and explicit layout hints are provided, making PosterCraft complementary to frameworks such as PosterMaker. Third, we plan to extend our unified pipeline beyond aesthetic posters to other text-heavy domains (e.g., infographics and dense scene text) by redesigning the data and text-specific rewards while still avoiding heavy modular architectures. Finally, we will push towards multilingual (Chinese/English and more) poster generation, where cross-language typography and layout bring additional challenges but also offer an important real-world testbed for the scalability and cross-cultural applicability of our framework.

### A.13 USAGE OF LLM

In preparing this manuscript, we employed large language models exclusively for editorial purposes, such as polishing grammar, enhancing readability, and shortening overly complex sentences. At no stage were ideas, methodologies, experiments, results, figures, or references generated by an

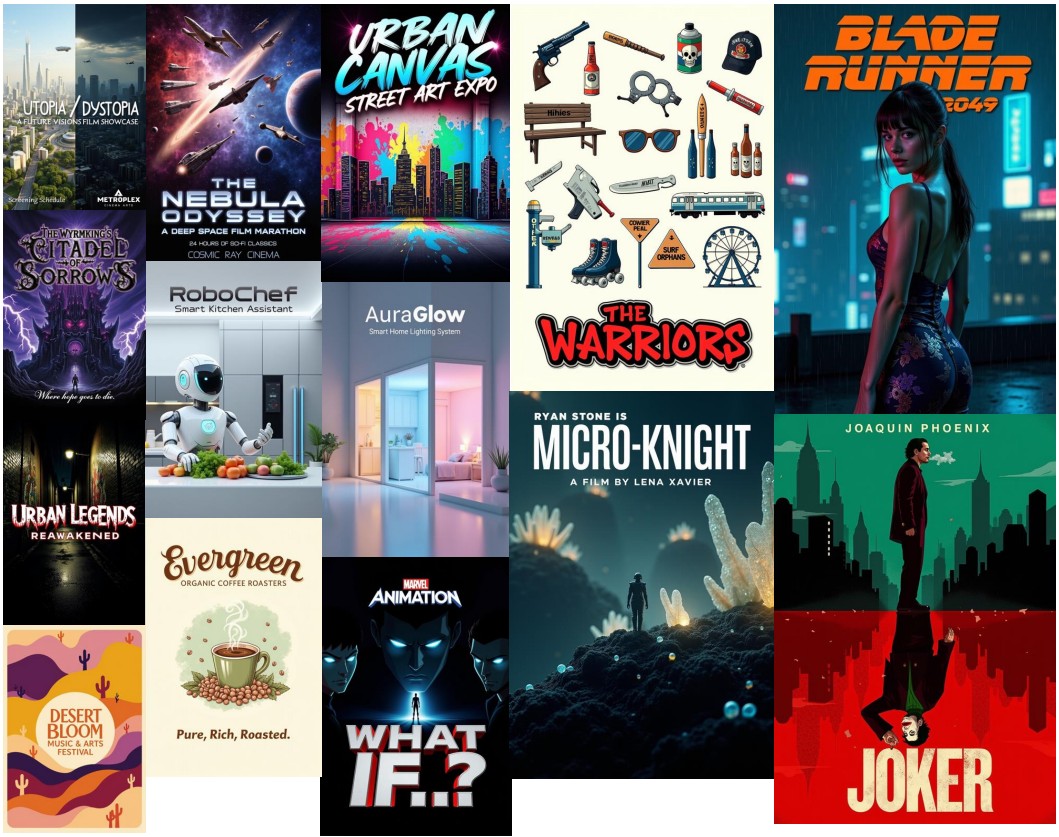

Figure 16: **Examples generated by our PosterCraft** demonstrating high diversity and aesthetic quality across themes including movies, product, and virtual reality. All generation results showcase genre-specific fidelity, text rendering, and layout aesthetic.

LLM, nor was unverifiable content introduced. All scientific contributions—including the design of the study, implementation of methods, execution of experiments, and validation of findings—were entirely carried out and verified by the authors, who retain full responsibility for the final content.

---

**Prompt A.1 (MLLM Scorer Prompt)**

Does this poster contain a large Billing Block or Credit Block at the bottom or *"4K ultrahd"* text at the top?
Based on your judgment, use the closest option to answer, and only return the label:
A. Yes. There is a large Billing Block or Credit Block at the bottom or *"4K ultrahd"* text at the top.
B. No. There is no Billing Block or Credit Block at the bottom and no *"4K ultrahd"* text at the top.

---

Figure 17: **Prompt** for MLLM Scorer in HQ-Poster-100K.

**Prompt A.2 (Gemini Caption Generation)**

Please write a structured and detailed caption in a single paragraph for this poster, covering the following five aspects in order:
**Poster Content**—Describe what is visually depicted.
**Poster Style**—Describe the visual or artistic tone, such as cinematic, surreal, minimalist, or other distinct aesthetics.
**Poster Text**—Provide the exact words shown in the image (title, subtitle, slogan, etc.) and their overall communicative intent.
**Text Style and Position**—Describe the typography in detail, including font style, size, texture, and how it visually blends or contrasts with the background (e.g., carved into a surface, embedded in light, wrapped by natural objects, etc.); also specify where each piece of text is positioned and its orientation angle in the frame.
**Layout**—Describe how the all elements are arranged to guide the viewer's focus.
Be specific, descriptive, and cohesive. Keep the response between 200 and 300 words, written as a single paragraph. Avoid listing or enumeration. Do not mention any design tools or generation methods. Write as if for a professional design catalog, highlighting how visual and typographic design choices form a unified and compelling narrative.

Figure 18: **Prompt** for Gemini Caption Generation in HQ-Poster-100K.

---

**Prompt A.3 (Gemini Mask Generation)**

Detect all text regions in the image. For each text region, provide its bounding box in `box_2d` format `[ymin, xmin, ymax, xmax]`. The coordinates for each bounding box must be a list of four integers `[ymin, xmin, ymax, xmax]`, normalized to the range `[0, 1000]`. Ensure the box completely covers the text area.

**MANDATORY GUIDELINES:**

- The `box_2d` coordinates `[ymin, xmin, ymax, xmax]` should be integers normalized to 0-1000.
- If no text is found in the image, the `"text_regions"` list in the JSON output should be empty.

**STRICT CONSTRAINTS:**

- Adhere strictly to the JSON output format specified below.
- Do not include any explanations, apologies, or conversational text outside of the JSON structure.
- Ensure the provided normalized coordinates are accurate.

**RESPONSE FORMAT:**

- Respond with a single JSON object. Do NOT use markdown (e.g., ` ```json ... ``` `).
- The JSON object must have a single key `"text_regions"`.
- The value of `"text_regions"` must be a list of `bounding_boxes`.
- Each `bounding_box` must be a list of four integer coordinates `[ymin, xmin, ymax, xmax]`, normalized to `[0, 1000]`.
- Example of the required JSON structure for `"text_regions"` containing two bounding boxes:

```
[
  [ymin1, xmin1, ymax1, xmax1],
  [ymin2, xmin2, ymax2, xmax2]
]
```

- The complete JSON object should look like this:

```
{
  "text_regions": [
    // List of bounding_boxes as shown above
    // e.g., [[20, 10, 50, 100], [70, 150, 100, 250]]
  ]
}
```

- If no text is found, the output should be: `{"text_regions": []}`.
- Provide ONLY this JSON object.

Now, based SOLELY on your comprehensive image analysis, provide ONLY the JSON object detailing all detected text regions and their normalized `box_2d` coordinates `[ymin, xmin, ymax, xmax]` as specified.

---

Figure 19: **Prompt** for Gemini Mask Generation in HQ-Poster-100K.

> **Prompt A.4 (Prompt Alignment Evaluation)**
>
> You are an expert in evaluating image content and font style against a given text prompt. You will be given an image and an original text prompt that was intended to generate an image similar to the one provided. Your task is to assess whether the image is substantially consistent with the original text prompt based on the criteria below.
> Original Text Prompt: "{original_prompt_text}"
> Evaluation Criteria (Prioritized):
>
> 1. **Text Accuracy:**
>     - Thoroughly analyze all text visible in the image. Check for any inaccuracies such as typos, missing characters/words, or extra characters/words when compared to the "Original Text Prompt". This is the MOST CRITICAL factor. If ANY such error is found, the decision MUST be "0".
>
> 2. **Text Style and Positioning:**
>     - If text is present, does its style (font, color, decoration) and positioning (layout, orientation) in the image reasonably align with what is described or implied in the "Original Text Prompt"?
>
> 3. **Overall Content, Artistic Style, and Visual Appeal:**
>     - Does the overall image content (subjects, scene, objects) and artistic style align well with the "Original Text Prompt"?
>     - Is the image generally clear, well-composed, and visually appealing in the context of the prompt?
>
> Output Format: Based on your assessment, output ONLY a JSON object in the following format: {{"final_decision": "1"}} if the image is substantially consistent with the original prompt across the prioritized criteria (especially if no text errors are found when text is intended) and should be kept. {{"final_decision": "0"}} if there are ANY discrepancies in Text Accuracy (typos, missing/extra characters/words), or significant issues in other critical criteria, or overall poor alignment, meaning the image should be discarded.
> Strict constraints:
>
> - Only output the JSON object.
> - Do NOT include any additional text, explanation, or markdown.
> - Use exactly "0" or "1" as the value for "final_decision".

Figure 20: **Prompt** for Prompt Alignment Evaluation in Poster-Preference-100K

---

**Prompt A.5 (Best-of-6 Selection)**

You are a professional Poster Designer. Your task is to evaluate six generated posters based on a design brief ("Original Prompt") and select the single best poster, or indicate if none are suitable.

**Evaluation Process:**

1. **Textual Accuracy (Paramount Importance):**
   - First, assess all posters for textual accuracy against the "Original Prompt". Text (if any is specified or implied by the brief) MUST be **perfectly accurate**:
     - No typographical errors.
     - No missing or extra characters/words.
   - **A poster with any textual flaw cannot be chosen as the best IF an alternative poster with perfect text exists.**

2. **Content Alignment and Aesthetic Value:**
   - This criterion is used to select among posters that have passed the textual accuracy check.
   - The chosen poster should:
     - Provide content as close as possible to the "Original Prompt".
     - Demonstrate the highest possible aesthetic value (considering composition, color palette, typography, imagery, and overall visual impact).

**Selection Logic:**

- **Ideal Case:** If one or more posters have **perfect textual accuracy**, select from THIS group the single poster that best meets Criterion 2 (Content Alignment and Aesthetic Value).

- **Special Case (All Posters Have Textual Flaws):** If ALL six posters have some textual inaccuracies, then no poster meets the primary standard for "best." In this situation, you MUST output "none".

- **Fallback Case (This should ideally not be reached if "Special Case" is handled correctly):** If the logic leads here unexpectedly after "Special Case" consideration, and no poster has perfect text, but a selection is still forced, choose the poster that, despite its textual flaws, is superior when evaluated SOLELY on Criterion 2 (Content Alignment and Aesthetic Value across all six flawed images). However, prioritize outputting "none" if all have text flaws.

Original Prompt (Design Brief): "`original_prompt`"
Select the image ("1", "2", "3", "4", "5", "6", or "none") that best meets these requirements.
Respond ONLY with a JSON object in ONE of these exact formats: {{"best_image": "1"}} OR {{"best_image": "none"}}
Strict constraints:

- Only output the JSON object.

- Do NOT include any additional text or markdown.

- Use exactly "1", "2", "3", "4", "5", "6", or "none" to refer to your selection.

---

Figure 21: **Prompt** for Best-of-6 Selection in Poster-Reflect-120K.

---

**Prompt A.6 (Feedback Collection)**

Internally compare the first poster against the second poster, focusing strictly on visual content layout and overall aesthetic style. Based on this internal comparison, provide detailed and specific suggestions in two aspects: 1. Poster Content Suggestions 2. Aesthetic Style Optimization Suggestions. Act as a professional poster designer. Deliver highly detailed, specific, and actionable feedback in the form of standardized image editing instructions.
MANDATORY GUIDELINES:

- The second poster must be fully followed as the standard. Identify and correct all visual layout and style discrepancies based on this reference.

- Focus exclusively on content and visual/aesthetic design. Completely ignore any issues related to text, typography, wording, spelling, rendering, or text styling.

STRICT CONSTRAINTS:

- NEVER mention the second poster, reference, or target.

- NEVER use comparative phrases such as "similar to the second poster" or "make it like the second poster".

- ONLY describe the editing instructions for Poster 1, framed as standalone improvement tasks.

RESPONSE FORMAT: Response should be formatted as clearly structured json schema: {`Poster Content Suggestions`: str, `Aesthetic style optimization suggestions`: str} Return ONLY the JSON object itself, without any introductory text or markdown formatting.

Figure 22: **Prompt** for Feedback Collection in Poster-Reflect-120K.

---

**Prompt  A.7 (OCR Evaluation)**

You are an OCR evaluation assistant. Follow these steps exactly on the attached image:

1. Ground-Truth Extraction (from the design prompt only):
   - Do NOT read text from the image for GT.
   - Parse ONLY the following design prompt and extract ALL text strings that should appear on the poster (titles, subtitles, dates, slogans, venue, etc.), preserving spaces and punctuation exactly: `original_prompt_text`
   - Order them in spatial sequence (top→bottom, left→right) and concatenate into **raw GT text**.

2. OCR Extraction (from the attached image):
   - Run OCR on the provided image and extract ALL rendered text exactly as it appears.
   - Preserve visual reading order (top-left→bottom-right). This is your **raw OCR text**.

3. Text Normalization (apply to BOTH raw GT and raw OCR before comparison):
   - Convert all letters to lowercase.
   - Remove ALL punctuation characters: `.,;:!?'"-()[]{}...`
   - Collapse any sequence of whitespace/newlines into a single space.
   - Trim leading and trailing spaces.

4. Character-Level Alignment & Error Counting:
   - Align the **normalized** GT text and OCR text **character by character**.
   - Count four categories:
     - **Insertion (I)**: extra character in OCR not in GT ("more").
     - **Deletion (D)**: GT character missing in OCR ("less").
     - **Substitution (S)**: OCR character differs from GT character ("render error").
     - **Correct match (C)**: identical characters.

5. Metrics Calculation:
   - Let N = total normalized GT characters = C + D + S.
   - Let P = total normalized OCR characters = C + I + S.
   - Let T = total compared characters = C + I + D + S.
   - **Character Accuracy** = C / T.
   - **Text Precision** = C / (C + I + S).
   - **Text Recall** = C / (C + D + S).
   - **Text F-score** = 2 * Precision * Recall / (Precision + Recall).

6. Final JSON Output (strictly this format, no extra keys or commentary):

```
{     "GT_text":   "<normalized GT text>",     "OCR_text":
"<normalized OCR text>",          "total_GT_chars":  N,
"correct_chars":  C,   "insertions":  I,   "deletions":
D,      "substitutions":  S,     "accuracy":  "XX.XX%",
"precision":  "YY.YY%",          "recall":  "ZZ.ZZ%",
"f_score":  "WW.WW%" }
```

Figure 23: **Prompt** for OCR Evaluation.

---

**Prompt A.8 (Preference Evaluation)**

Your task is to evaluate a single input image containing two sub-images side-by-side (Left: L, Right: R), both generated from the "Original Prompt". Compare them on Aesthetic Value, Prompt Alignment, Text Accuracy, and Overall Preference.

**General Evaluation Protocol:** For each of the four categories:

1. Provide a brief textual analysis justifying your choice.

2. Make a definitive choice: "L" (Left is superior), "R" (Right is superior), or "none".

**When to Choose "none":** You **must select "none"** for a category if:

a) L and R are tied or indistinguishable in quality for that category.

b) The category is not applicable.

c) After careful review, you cannot definitively determine a superior side.

d) **Crucially: L and R exhibit clear, offsetting strengths and weaknesses *within that specific category*. If L excels in one aspect of the category while R excels in another, and these trade-offs make declaring an overall winner for that category difficult or misleading, choose "none". Do not attempt to weigh these distinct, offsetting pros and cons to force a preference.**

Your careful judgment is vital.
Original Prompt: "`original_prompt`"
Please provide your evaluation in the JSON format specified below.

1. **Aesthetic Value:**

   - Evaluate visual appeal: harmony and consistency of background style, text style (if present), thematic consistency between background/text, overall content/text layout, and how the artistic style (background, content, text) aligns with the "Original Prompt".
   - `aesthetic_value_explanation`: Your brief analysis.
   - `aesthetic_value`: Choose "L", "R", or "none" (if L/R are equally pleasing/coherent, a choice is impossible, or they exhibit offsetting aesthetic strengths/weaknesses as per the "When to Choose 'none'" protocol).
   - Respond with: `{{"aesthetic_value": "L/R/none", "aesthetic_value_explanation": "Your analysis..."}}`

2. **Prompt Alignment (excluding text elements and artistic style):**

   - Evaluate how well non-textual elements (subjects, objects, scene) in L and R match the "Original Prompt".
   - `prompt_alignment_explanation`: Your brief analysis.
   - `prompt_alignment`: Choose "L", "R", or "none" (if L/R align equally well/poorly, it's too close to call, or they exhibit offsetting strengths in alignment as per the "When to Choose 'none'" protocol).
   - Respond with: `{{"prompt_alignment": "L/R/none", "prompt_alignment_explanation": "Your analysis..."}}`

Figure 24: **Prompt** for Preference Evaluation.

---

**Prompt A.9 (Preference Evaluation)**

1. **Text Accuracy (if applicable):**
   - Evaluate text in L and R based *only* on textual content specified/implied in the "Original Prompt".
   - Focus *only* on:
     - **Accuracy:** All prompt-specified words/characters present, no typos/misspellings/alterations?
     - **Recall:** All intended textual elements from prompt included? Any missing words/phrases?
   - **Ignore text style, font, visual appeal, legibility (unless it prevents determining accuracy/recall), and placement.**
   - `` `text_accuracy_explanation` ``: Your brief analysis.
   - `` `text_accuracy` ``:
     - First, determine if "none" is appropriate (as per the general "When to Choose 'none'" protocol, especially if L is better on Accuracy but R on Recall, or vice-versa; or if performance is identical/text N/A).
     - If "none" is not chosen, select "L" if L is demonstrably superior overall in combined text accuracy and recall, or "R" if R is.
   - Respond with: `{{"text_accuracy": "L/R/none", "text_accuracy_explanation": "Your analysis..."}}`

2. **Overall Preference:**
   - Considering all above aspects (aesthetics, alignment, text accuracy) and any other factors relevant to the "Original Prompt".
   - `` `overall_preference_explanation` ``: Your brief analysis.
   - `` `overall_preference` ``: Choose "L", "R", or "none" (if L/R are equally preferred, a choice is impossible, or they present compelling but different and offsetting strengths across categories making neither holistically superior, as per the "When to Choose 'none'" protocol).
   - Respond with: `{{"overall_preference": "L/R/none", "overall_preference_explanation": "Your analysis..."}}`

Respond ONLY with a single JSON object in the following format:
```
{ "aesthetic_value": "your_choice_for_aesthetic",
"aesthetic_value_explanation": "Your brief analysis for
aesthetics...",
"prompt_alignment": "your_choice_for_alignment",
"prompt_alignment_explanation": "Your brief analysis for
prompt alignment...",
"text_accuracy": "your_choice_for_text",
"text_accuracy_explanation": "Your brief analysis for text
accuracy...",
"overall_preference": "your_choice_for_overall",
"overall_preference_explanation": "Your brief analysis for
overall preference..."
```
} Replace placeholders with your choices ("L", "R", "none") and analyses.
Strict constraints:

- Only output the JSON object.
- No additional text or markdown.
- Each choice value (e.g., "aesthetic_value") must be "L", "R", or "none".
- Explanation fields must contain your textual analysis.

Figure 25: **Prompt** for Preference Evaluation.

**Prompt A.10 (Final Retouching Instructions)**

**Character**
You are a professional image retouching artist tasked with finalizing a single retouching approach based on the user's preferences and previous proposals. Your expertise ensures that the final approach integrates key aspects from different suggestions or follows a single selected approach in full.

**Background**
The user has reviewed previous retouching approaches and provided feedback or specific instructions for a final retouching plan that aligns with their creative goals.

**Ambition**
Your goal is to either choose one of the previously proposed approaches that best matches the user's vision or create a new, cohesive retouching approach by combining elements from different suggestions. Ensure that the final approach fully respects the user's instructions and creative intent.

**User Instruction**
User says: *"{user_instruction}"*

**Task**

1. Review the provided retouching approaches and the user's feedback or instructions.

2. Decide whether to:
    - Select a single approach that fits the user's description.
    - Create a new approach that integrates relevant aspects from different suggestions.

3. Final Approach:
    - Describe the adjustments to **Light** (exposure, contrast, highlights, shadows, blacks, and whites) and **Color** (temperature, tint, vibrance, and saturation).
    - For each adjustment, specify **which objects or areas of the image** are most affected and describe the specific details (e.g., "the intricate carvings on the roof are highlighted by a gentle increase in exposure").
    - Explain the expected **visual effect on these objects**, such as "the water reflections appear richer and more defined" or "the sky becomes softer and more inviting."
    - For each **individual HSL adjustment** (Red, Orange, Yellow, Green, Cyan, Blue, Purple, Magenta), explain why it is necessary and describe the expected visual change for specific objects (e.g., "the red tones in the window frames become more vivid to emphasize their ornate design").
    - Organize the description as a step-by-step plan, indicating the sequence of adjustments.

**Guidelines for Description:**

- Avoid providing exact numerical values—focus on explaining how the adjustments affect the image's visual presentation.
- Mention specific objects, areas, and their corresponding changes to help visualize the effect.
- Ensure the approach remains detailed, logical, and cohesive, and does not exceed 100 words.

Figure 26: **Prompt** for Final Retouching Instructions.

Table 6: Effect of applying the full PosterCraft pipeline to SD3.5. Text F–score and HPSv3 are computed on our aesthetic text–to–poster prompts. The Gemini win rate vs. SD3.5 reflects overall preference between SD3.5 with our training pipeline and the vanilla SD3.5 baseline.

| Model variant | Text F-score ↑ | HPSv3 ↑ | Win rate vs. SD3.5 ↑ |
|---|---|---|---|
| SD3.5 (baseline) | 0.479 | 10.122 | 50.0 |
| SD3.5 + full PosterCraft (4 stages) | **0.554** | **10.330** | **58.1** |

## A.14 Generalizability beyond Flux-dev: Experiments on SD3.5

To verify that our workflow is not tied to Flux-dev, we further apply the same four-stage post-training pipeline (text rendering → poster SFT with region-aware calibration → aesthetic-text preference RL → VLM reflection) to Stable Diffusion 3.5-Medium (SD3.5). We keep the SD3.5 architecture and sampler unchanged and only reuse our datasets and method designs.

Figure 27 presents a visual comparison on several poster prompts: the left column shows raw SD3.5 outputs, and the right column shows results after our PosterCraft pipeline. The optimized posters exhibit sharper and more consistent text, clearer layout structure, and improved global aesthetics, while preserving the semantics of the original prompts.

Quantitatively, Table 6 reports text F-score on our text-to-poster prompts, the HPSv3 aesthetic score, and the Gemini win rate when comparing SD3.5 with and without our post–training. Our pipeline brings consistent gains on all three metrics, mirroring the trends observed on Flux-dev. This confirms that the proposed framework is architecture-agnostic and can serve as a general post-training recipe for modern diffusion backbones, rather than a Flux-specific trick.

## A.15 Why a Unified PosterCraft Framework Works

Our goal is not to propose a new backbone, but to show that a single diffusion model can learn text, layout, and visual style for posters through a carefully staged, unified workflow. Traditional poster systems typically decompose these factors into separate networks or ControlNet-style branches (e.g., independent text, layout, and style modules), which increases inference cost and makes global optimization difficult. PosterCraft instead keeps one backbone and upgrades it via four stages that are all defined on simple image–text pairs.

This unified view is feasible because each stage is poster-specific yet complementary: (i) a text-rendering stage first calibrates stroke-level accuracy and small-font legibility on synthetic Text-Render-2M without changing the architecture; (ii) a region-aware SFT stage on HQ-Poster-100K rebalances losses over major text, minor text, and non-text regions so that the model learns realistic poster layouts and typography while preserving the strengthened text ability; (iii) an aesthetic–text preference RL stage optimizes a single preference signal that jointly measures aesthetic quality and text correctness, instead of focusing on only one of them; and (iv) a customized VLM-reflection stage feeds critique embeddings back into the prompt space, further refining layout and style without adding heavy controllable signals or liking a general editing models.

Because all four stages share the same backbone and operate on unified supervision, PosterCraft learns to trade off text, layout, and style within one model rather than across loosely coupled modules. Empirically, we observe consistent gains over the base FLUX and SD3.5 backbones on both poster benchmarks and generic text-rendering datasets, suggesting that the unified framework is not tied to a particular architecture and can serve as a general post-training recipe for high-quality, text-aware image generation.

## A.16 Dataset Availability and Statistical Analysis

**Public availability.** As clarified in the main paper, all datasets in this work are automatically constructed from publicly available imagery rather than private or sensitive data. Text-Render-2M is built by synthetically rendering diverse texts onto high-quality real images drawn from publicly available image datasets (natural scenes, products, artworks, historical photos, etc.). HQ-Poster-100K is collected from publicly available poster resources with automatic filtering and text-mask

generation. We will release: (i) the complete Text-Render-2M dataset (rendered image + prompt), (ii) the full HQ-Poster-100K poster images with prompts and text masks, and (iii) the PosterCraft model checkpoints trained on these data, under a non-commercial research license. HQ-Poster-100K contains third-party copyrighted materials and is released strictly for non-commercial research under a fair-use principle; we do not claim ownership of these materials, and downstream users are responsible for complying with applicable copyright regulations.

### A.16.1 STATISTICS OF TEXT-RENDER-2M

Text-Render-2M is a synthetic data of rendered texts on real photographic backgrounds. We compute corpus-level and scene-level statistics over all rendered samples.

Table 7: Statistics of Text-Render-2M (rendered texts + background images).

| Aspect | Category | Portion |
|---|---|---|
| Text instances per image | 1 instance | 43% |
| | 2 instances | 36% |
| | 3 instances | 21% |
| Text content type | Template-based grammatical phrases | 71% |
| | Random alphanumeric strings | 29% |
| Text length (words per instance) | 1–5 words / very short strings | 23% |
| | Short phrases ($\approx$5–10 words) | 47% |
| | Sentence-level text ($\approx$10–20 words) | 22% |
| | Long sentences or short paragraphs ($>$20 words) | 8% |
| Orientation / layout | Horizontal | 61% |
| | Vertically rotated ($90^\circ$) | 17% |
| | Vertically stacked (one character per line) | 22% |
| Background scene type (underlying images) | Macro nature & ecology (insects, plants, fungi) | 21% |
| | Everyday outdoors & landscapes | 18% |
| | Urban scenes & architecture | 16% |
| | People & portraits | 13% |
| | Products & still-life | 14% |
| | Artistic paintings & illustrations | 10% |
| | Historical / archival / documentary photos | 8% |

These statistics show that Text-Render-2M covers multi-instance layouts, several orientation modes, and a substantial share of sentence- and paragraph-level text, while the photographic backgrounds span diverse natural, urban, artistic, and historical scenes rather than being limited to simple natural images.

### A.16.2 STATISTICS OF HQ-POSTER-100K

Unlike the synthetic Text-Render-2M data, HQ-Poster-100K consists of real-world aesthetic posters. Most posters revolve around a main title, an optional tagline, and occasionally compact auxiliary information.

Taken together, these statistics highlight the complementary roles of the two datasets in our training pipeline. Text-Render-2M provides controlled diversity in text length, orientation, and cross-domain photographic backgrounds, which is crucial for robust text rendering. HQ-Poster-100K, in turn, supplies fully designed aesthetic posters where layout structure, visual style, and text–image integration are jointly optimized by human designers—exactly the style and layout distribution that the later stages of PosterCraft are trained to model.

Table 8: Text statistics of HQ-Poster-100K.

| Aspect (Text) | Category | Portion | Notes |
|---|---|---|---|
| Text instances per poster | 1–2 blocks | ≈48% | Mostly only the main title. |
| | 3–4 blocks | ≈36% | Title plus one or two short taglines or secondary blocks. |
| | >4–5 blocks | ≈16% | Additional taglines or other information. |
| Text content type | Titles / named entities | ≈46% | Titles, character names, locations, etc. |
| | Taglines / short phrases | ≈34% | Promotional slogans, mood-setting phrases. |
| | Others | ≈20% | Cast lists, logos, copyright lines, release information. |
| Text length (per block) | 1–10 words (short) | ≈72% | Short titles and concise taglines. |
| | 10–20 words (medium) | ≈21% | Longer slogans or brief descriptions. |
| | >20 words (long) | ≈7% | Mainly dense billing blocks at the bottom. |

Table 9: Visual and aesthetic statistics of HQ-Poster-100K.

| Aspect (Visual) | Category | Portion | Key descriptors |
|---|---|---|---|
| Visual style / art direction | Cinematic / realistic | ≈63% | Dramatic lighting, clear depth of field, photographic look. |
| | Illustrative / graphic | ≈24% | Hand-drawn art, comic/graphic-novel styles, painterly or pop-art treatments. |
| | Minimalist / symbolic | ≈13% | Strong negative space, silhouettes, emblematic icons. |
| Subject matter / composition | Character portrait | ≈43% | One or two close-up faces, expressive emotions. |
| | Ensemble / group composition | ≈26% | Multiple characters in montage or tiered layouts. |
| | Action or scene-driven | ≈19% | Dynamic poses, chase scenes, large landscapes or cityscapes. |
| | Object- / concept-centric | ≈12% | Key prop or symbolic object (helmet, device, broken glass, etc.). |

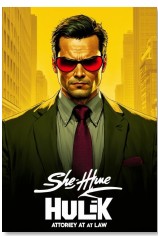 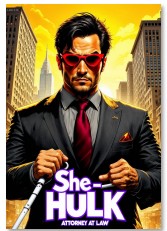

This vertical poster for *She-Hulk: Attorney at Law* features a stylized image of Charlie Cox as Matt Murdock, a legally blind man wearing a dark gray suit, a light gray shirt, and a maroon tie, holding a white cane with both hands at his chest. He is wearing his iconic red-lensed sunglasses, looking cool and confident, with the sunlit silhouette of Los Angeles buildings behind him. The poster has a sunny, stylized look with saturated yellow lighting and a slightly textured effect. The title *"She Hulk"* is displayed prominently in stylized white text with a white border, positioned in the bottom right quadrant of the poster, with "She" in a smaller script font positioned above "Hulk" which is rendered in a larger, bold, sans-serif font. Directly below "Hulk", in a smaller, all-caps, sans-serif font, is the subtitle *"ATTORNEY AT LAW,"* creating a stacked visual hierarchy. Across the top of the poster, in a large, dark purple, all-caps, sans-serif font, is the text *"CHARLIE COX IS MATT MURDOCK."* All text is horizontally oriented. The composition emphasizes Matt Murdock in the foreground, centered within the frame, with the cityscape providing a dynamic backdrop, and the title treatment placed in the bottom right to balance the weight of the subject and text at the top. The overall layout draws the eye from the actor's name and character title at the top, down to his portrait and the film's title at the bottom, creating a clear flow for the viewer.

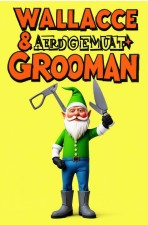 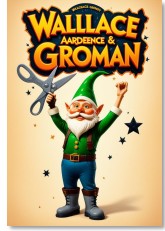

This poster features a whimsical, stop-motion animated style, portraying a gleeful garden gnome in a vibrant green outfit, blue pants, and gray boots with white cuffs, holding a large pair of open garden shears above his head against a warm, pale beige background. The text includes the title *"Wallace & Gromit"* in large, orange, blocky, three-dimensional lettering with a subtle darker outline and highlight, appearing at the top of the poster and angled slightly downward from left to right. Below the title, in a smaller, jagged-edged font, the subtitle *"VENGEANCE MOST FOWL"* is displayed in bright yellow, positioned centrally and spanning the width of the title. Above the main title, in a smaller black, sans-serif font, the word *"AARDMAN"* is positioned slightly to the left, accompanied by a small black star. The overall layout positions the text prominently at the top, drawing the eye with its bold colors and distinctive typography, while the charming character and oversized shears are centered below, creating a balanced composition that is both playful and slightly ominous, hinting at the film's blend of humor and mystery.

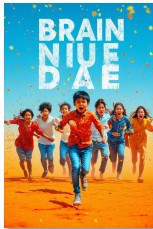 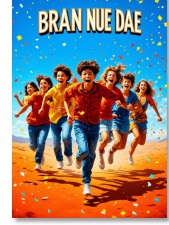

This vibrant poster features a group of seven people, mostly adults and one boy, captured mid-stride on a warm, dusty orange landscape under a bright blue sky filled with scattered white and yellow confetti. They are all facing forward, expressions ranging from joyous exuberance to determined intensity, suggesting a journey or collective movement. The style is upbeat and celebratory, with the clear blue sky and warm tones conveying a sense of optimism and adventure. The central text, *"BRAN NUE DAE"*, appears in large, light blue, distressed block letters that have a textured, slightly worn appearance, hinting at a lived-in, authentic story. This title is angled slightly upward from left to right and dominates the upper two-thirds of the frame, positioned prominently against the vast blue sky. The figures below are arranged horizontally across the lower third of the poster, their poses and expressions drawing the eye towards the lively scene. The overall layout creates a dynamic composition, with the bold title acting as an anchor at the top and the active figures below providing the human element, inviting the viewer to join them on their "new day" adventure.

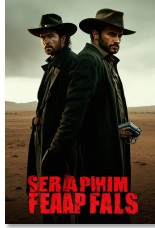 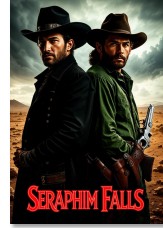

The poster for "Seraphim Falls" features two men, one in a dark coat and hat, the other in a green shirt with a visible holster and pistol, standing back-to-back in a vast, arid landscape under a dramatic, cloudy sky; the scene evokes a sense of the Western genre and intense conflict, conveying a rugged and serious tone through its cinematic style. The prominent text at the bottom of the poster reads *"SERAPHIM FALLS,"* serving as the film's title. The title is presented in a bold, distressed, red sans-serif font, appearing textured and gritty, as if weathered by the harsh environment depicted. The text is horizontally oriented and centered at the bottom of the frame. The overall layout places the two figures prominently in the mid-ground, dominating the upper and central portions of the poster, while the title anchors the composition at the base, drawing the viewer's eye downwards after taking in the characters and setting. The stark contrast between the dark figures and the lighter, dusty landscape, combined with the intense color and texture of the title, creates a visually striking and impactful design that strongly communicates the film's Western theme and dramatic nature.

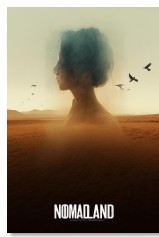 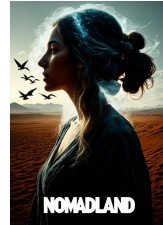

The poster for "Nomadland" features a layered image, with a portrait of a woman, likely the protagonist, superimposed over a vast, arid landscape. The visual style is muted and contemplative, evoking a sense of quietude and the stark beauty of the American West. The poster text consists solely of the film's title, *"Nomadland,"* presented in a classic serif font. The letters are white, providing a strong contrast against the earthy tones of the foreground, and appear to be positioned horizontally at the bottom of the frame, subtly grounded within the landscape. The overall layout draws the eye upwards from the prominent title, through the expansive scenery, and finally to the expressive portrait of the woman, creating a visual narrative that connects the individual to the sweeping environment she inhabits. The ethereal quality of the overlaid portrait and the presence of birds soaring against the sky further enhance the film's themes of freedom and transience within a grand, natural setting.

Figure 27: **Visual comparison of poster generation performance.** The left column displays the raw outputs from the original **SD3.5-Medium**. The right column presents the results optimized via our **PosterCraft** pipeline. The text boxes on the right contain the complete input prompts used for generation. For a fair comparison, the random seed is fixed at 0 for all inference steps.

*"The whimsical poster for the fantasy film 'Moonpetal Grove' depicts a young girl with luminous blue eyes reaching out to touch a giant, glowing moonpetal flower in an enchanted forest at twilight. The forest is filled with oversized, fantastical flora, faintly glowing mushrooms, and sparkling fireflies, with silhouetted, ancient trees in the deep blue background. The artistic style is richly illustrative and magical, using a palette of deep blues, purples, and bioluminescent greens and yellows to create a dreamlike atmosphere. The film's title, 'Moonpetal Grove,' is rendered in an ornate, flowing, silver script font, with delicate, leafy tendrils entwining the capital letters. The title is arched gracefully across the top of the poster, positioned horizontally. Beneath it, the tagline 'Where magic takes root' is written in a smaller, whimsical gold serif font, also horizontal. The layout guides the viewer's eye from the enchanting title down to the central interaction between the girl and the magical flower. The glowing elements and intricate details invite exploration, promising a journey into a world of wonder and enchantment, perfectly captured by the fantastical typography."*

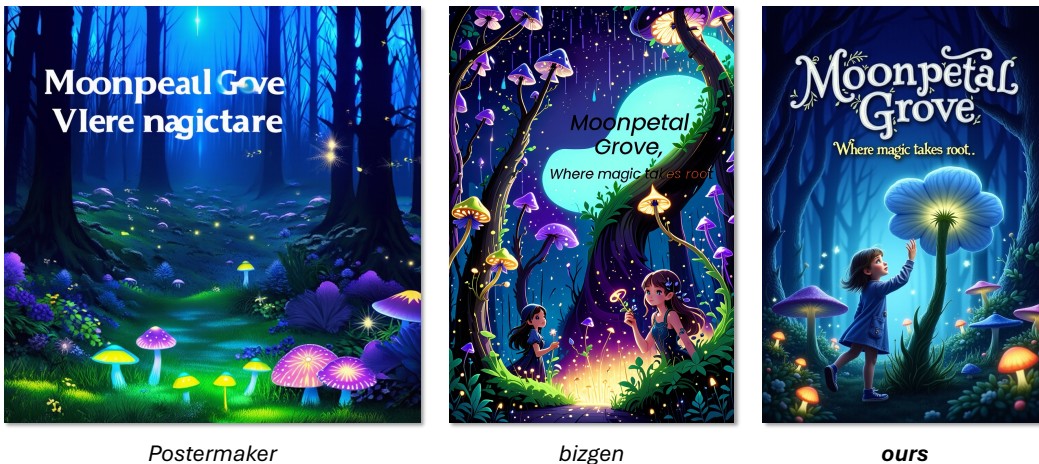

Postermaker       bizgen       **ours**

Figure 28: **Qualitative comparison against layout based methods (Part 1).** We present generation results from Postermaker (left), Bizgen (middle), and our **PosterCraft** (right). The corresponding creative briefs are displayed in the red boxes above each example. As illustrated, PosterCraft demonstrates superior performance in strict text rendering, layout arrangement, and aesthetic alignment with the prompt, whereas baseline models frequently exhibit text hallucinations or fail to capture the specified atmosphere.

*"This poster for the neo-western 'Dust Devil's Due' features a lone, silhouetted cowboy on horseback, riding into a fiery, dust-choked sunset over a desolate desert landscape. Jagged rock formations frame the horizon. The artistic style is stark and cinematic, using a dramatic color palette of deep oranges, reds, and shadowy blacks to create a sense of rugged isolation and impending conflict. The film's title, 'DUST DEVIL'S DUE,' is rendered in a bold, heavily distressed, and spurred serif font, reminiscent of classic Western typography. The letters are a weathered, sandy yellow and appear to be branded or burned into a dark, leather-like strip that runs horizontally across the bottom of the poster. The title is slightly arched. The tagline, 'Vengeance rides a pale horse,' is etched in a smaller, sharper, dark red font above the title. The layout is epic and atmospheric, with the vast desert and dramatic sky dwarfing the lone rider, emphasizing his solitary journey. The rugged, impactful title treatment at the bottom grounds the image and immediately establishes the film's genre and tone."*

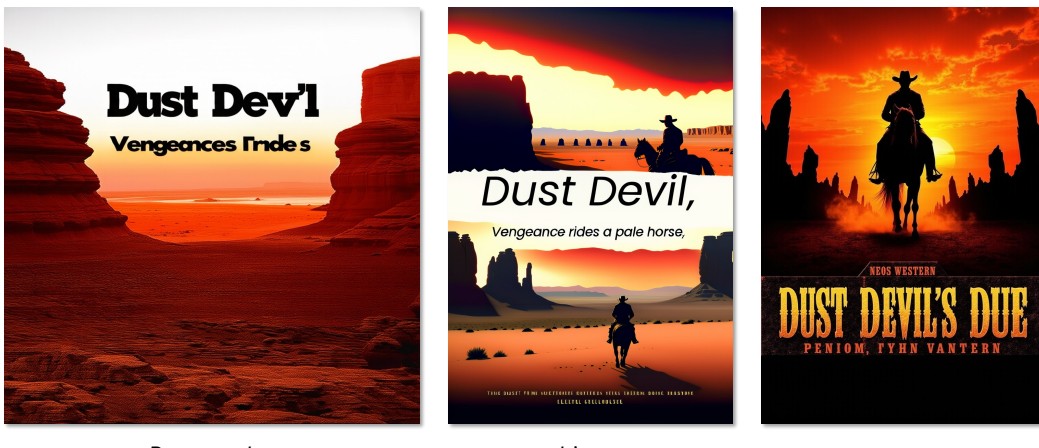

Postermaker       bizgen       **ours**

Figure 29: **Qualitative comparison against layout based methods (Part 2).**

*"A post-apocalyptic survival film poster features a lone survivor, heavily cloaked, trudging through a snow-covered, ruined city, a loyal dog by their side. The sky is a perpetual, toxic yellow-grey. The style is desolate and gritty, emphasizing hardship and resilience, with a muted, cold color palette. The title, \"THE LONG WINTER,\" is rendered in a stark, blocky, ice-blue sans-serif font with a cracked texture, positioned horizontally at the very bottom, almost buried in the snow. The actor's name, \"ANYA TAYLOR-JOY,\" is subtly placed in a small, white, weathered font at the top. The layout conveys a sense of vast emptiness and the arduous journey of the protagonist, with the icy typography reflecting the harsh environment."*

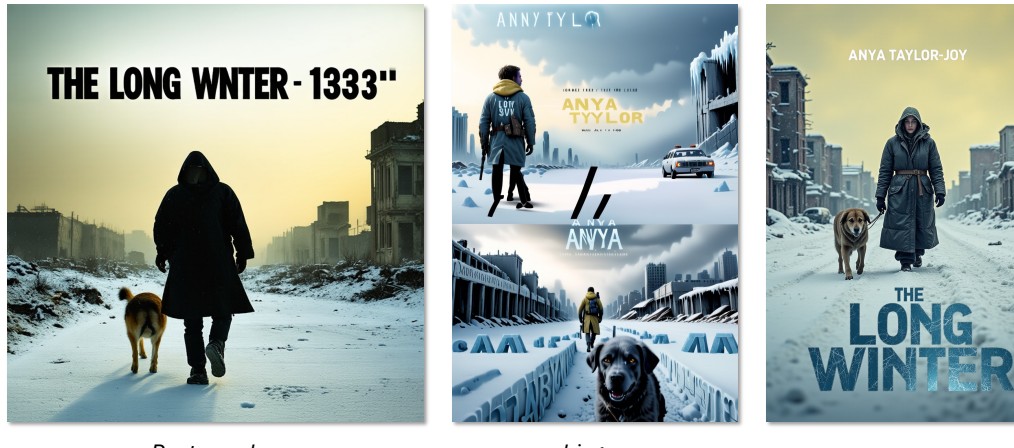

Figure 30: **Qualitative comparison against layout based methods (Part 3).**

*"This epic historical adventure poster shows a fleet of Viking longships sailing through a stormy, dark sea towards a rugged, mist-shrouded coastline. The style is dramatic and painterly, with a sense of rugged power and impending conflict, using a palette of dark blues, greys, and flashes of lightning. The title, \"NORTHWIND SAGA,\" is rendered in a heavy, runic-inspired, metallic silver font with a weathered texture, positioned horizontally at the bottom, appearing forged and ancient. The names of the lead actors, \"ALEXANDER SKARSGÅRD\" and \"NICOLE KIDMAN,\" are in a smaller, classic serif font above the title. The typography strengthens the film's Norse mythological and historical themes."*

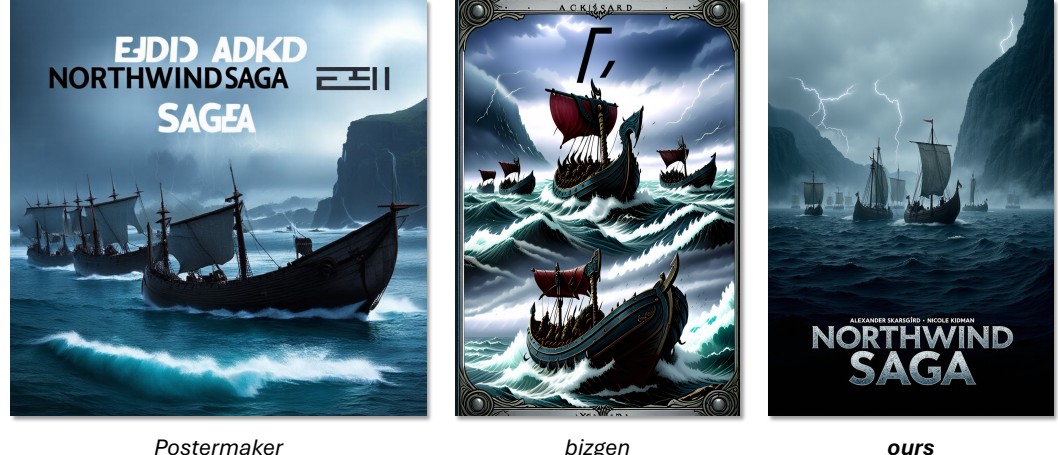

Figure 31: **Qualitative comparison against layout based methods (Part 4).**

*"A romantic drama poster features a silhouette of a couple embracing against a vibrant, abstract watercolor background of swirling pinks, purples, and blues, representing their emotional world. The style is artistic and emotive, focusing on the feeling of love rather than realistic depiction. The film's title, \"COLORS OF THE HEART,\" is written in an elegant, flowing, white script font, positioned diagonally across the center of the watercolor swirl, blending with the artwork. The tagline, \"Love paints its own picture,\" is in a smaller, delicate serif font below. The typography is soft and romantic, perfectly complementing the abstract, emotional visuals."*

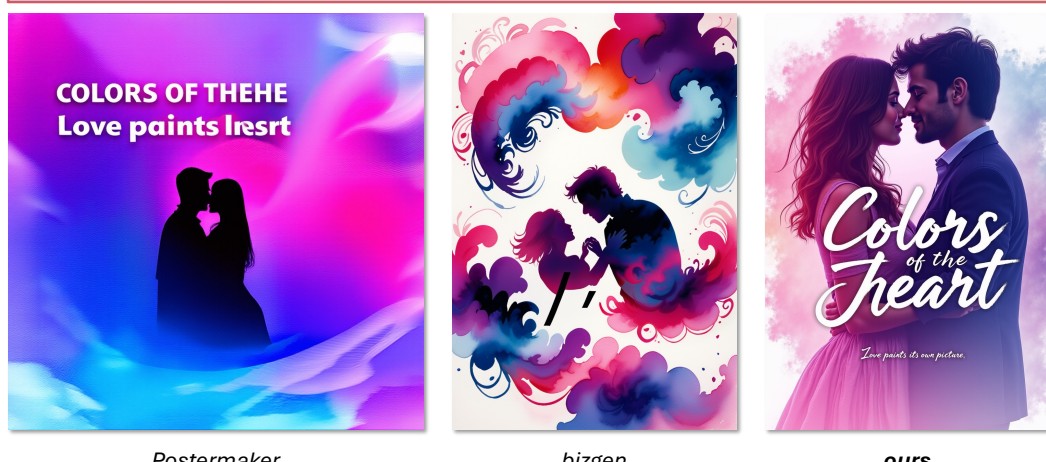

| Postermaker | bizgen | **ours** |

Figure 32: **Qualitative comparison against layout based methods (Part 5).**

*"This advertisement for 'Oasis Infusions' botanical gin features a sleek, clear glass bottle of gin, with delicate illustrations of juniper berries, citrus peel, and elderflower visible on the label, surrounded by fresh botanicals and ice cubes in a crystal glass. The background is a softly lit, deep teal. The style is sophisticated, refreshing, and artisanal, emphasizing natural ingredients and premium quality. The brand name, 'Oasis Infusions', is in a refined, contemporary serif font in a subtle silver, positioned horizontally at the top. The tagline 'A Sip of Serenity.' is in a smaller, elegant script font in white, centered below the bottle and glass. 'Crafted with Wild Botanicals' is discreetly printed on the bottle label. The layout is elegant and visually appealing, with the gin bottle and cocktail as the clear focus, and the sophisticated typography reinforcing the brand's premium, nature-inspired identity."*

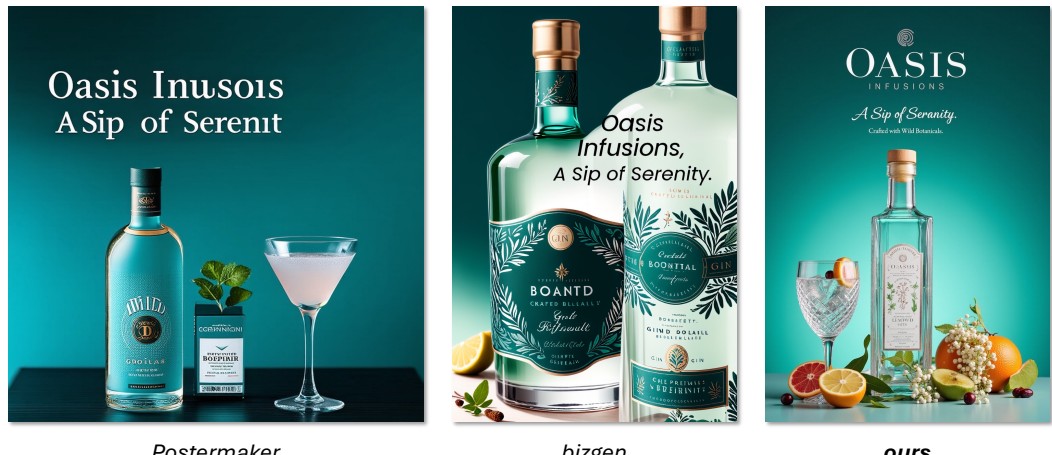

| Postermaker | bizgen | **ours** |

Figure 33: **Qualitative comparison against layout based methods (Part 6).**

*"This poster for 'The Ember & Grove' Fall Harvest Festival depicts a charming, rustic scene: a wooden cart overflowing with pumpkins, apples, and gourds, set against a backdrop of golden autumn trees and a distant farmhouse. Warm, dappled sunlight filters through the leaves. The style is warm, inviting, and traditional, celebrating the bounty of autumn. The festival title, 'Ember & Grove', is in a friendly, hand-lettered, slightly whimsical serif font in a rich burgundy, arcing above the cart. 'Fall Harvest Festival' is in a smaller, rustic sans-serif in dark orange, beneath the title. 'Live Music | Artisan Crafts | Fresh Produce | Oct 15-16 | Willow Creek Farm' is at the bottom in a clean, brown sans-serif. The layout is picturesque and abundant, with the harvest cart as the centerpiece, and the charming typography enhancing the festival's warm, community-focused atmosphere."*

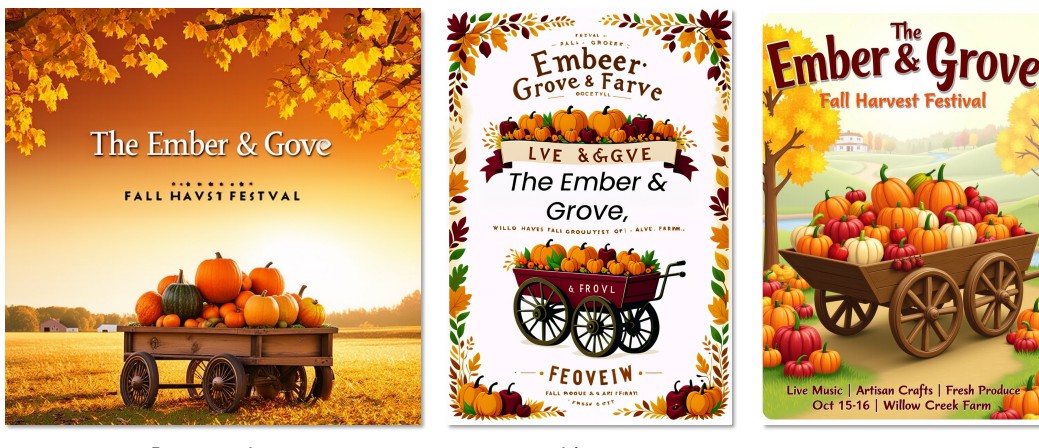

Postermaker         bizgen         **ours**

Figure 34: **Qualitative comparison against layout based methods (Part 7).**

*"This poster for the 'Crimson Tide Surf Competition' showcases a dramatic, action-packed photograph of a surfer riding a massive, curling wave, silhouetted against a fiery red and orange sunset. Water spray is frozen in mid-air. The style is exhilarating, powerful, and adventurous, capturing the raw energy of surfing. The competition title, 'CRIMSON TIDE', is in a bold, dynamic, sans-serif font with a slight wave-like distortion, rendered in a stark white that cuts through the vibrant sunset colors, positioned horizontally across the top third of the poster. 'Annual Surf Competition' is in a smaller, sharp sans-serif in yellow, beneath the title. 'Pipeline Beach | October 5-7 | Prizes & Glory!' is at the bottom in a clean white sans-serif. The layout is impactful, with the surfer and wave dominating the visual field, conveying the intensity of the sport, while the energetic typography enhances the event's thrilling and competitive nature."*

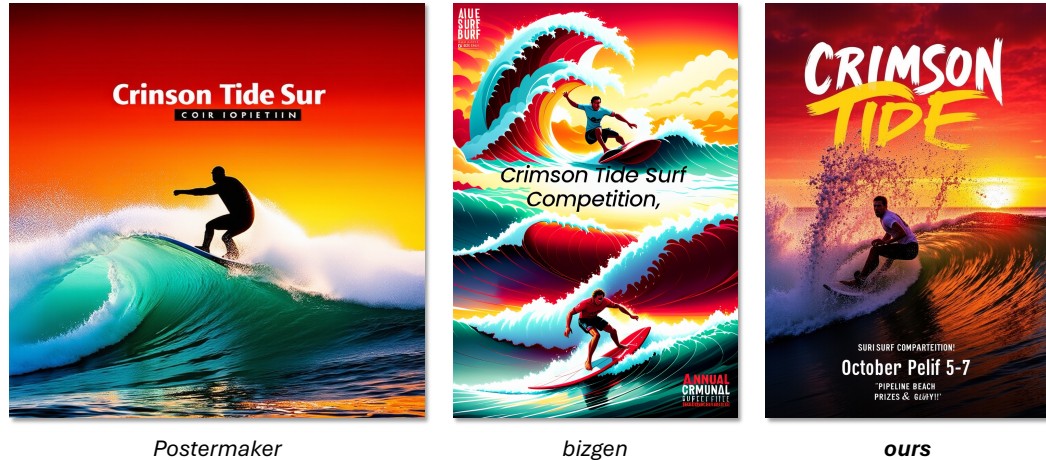

Postermaker         bizgen         **ours**

Figure 35: **Qualitative comparison against layout based methods (Part 8).**

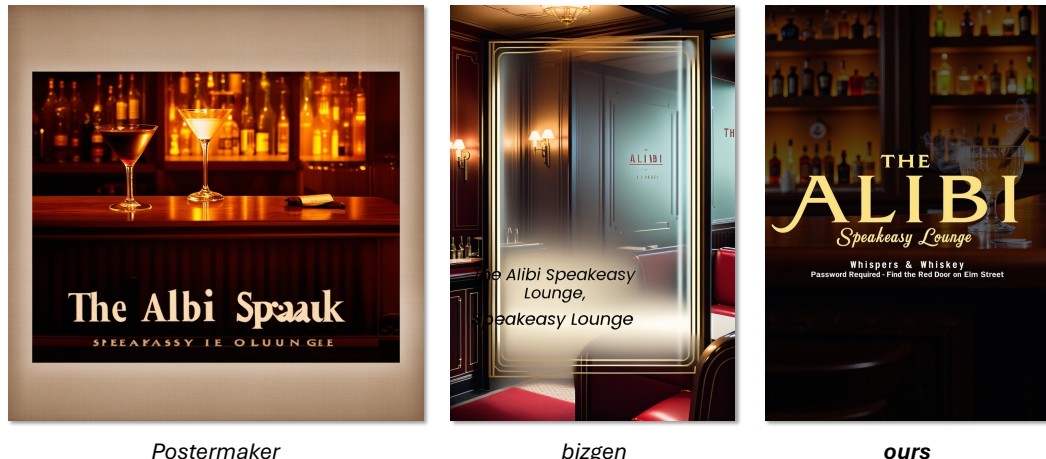

Figure 36: **Qualitative comparison against layout based methods (Part 9).**

