# OpenReview forum: "PosterCraft: Rethinking High-Quality Aesthetic Poster Generation in a Unified Framework"
_ICLR.cc/2026/Conference — ICLR 2026 Poster_

### Official Review · Reviewer_rUYo · 2025-10-26

**Soundness:** 2
**Presentation:** 3
**Contribution:** 2
**Rating:** 4
**Confidence:** 5

**Summary:**

This work proposes an unified diffusion-based framework for high-quality aesthetic poster generation. Authors design an end-to-end optimization workflow to jointly learn text rendering, artistic layout, and stylistic coherence.

**Strengths:**

This work achieve visually coherent results by integrating text, layout, and artistic style in a single generative process.
Authors proposes a high-quality Text-Render-2M dataset.
PosterCraft outperform all open-source baselines (Flux1.dev, SD3.5, BAGEL, etc.) and nearly match the closed-source Gemini-2.0-Flash-Gen.

**Weaknesses:**

The motivation for adopting a unified framework appears questionable. Visual styles and textual elements often have fundamentally different requirements, visual styles typically emphasize global consistency, whereas text demands attention to fine-grained details and stroke-level accuracy.How does the proposed method reconcile these competing objectives?

The paper lacks sufficient comparison with recent academic works, such as Postermaker[1]. It is recommended that the authors provide a more detailed discussion of how their approach differs from or improves upon these recent methods to better highlight the novelty and contribution of this work.
[1] Postermaker: Towards high-quality product poster generation with accurate text rendering

While the dataset scale is impressive, the lack of public availability and insufficient statistical analysis limit its value to the research community.

**Questions:**

Generating highly realistic text in an end-to-end manner remains a significant challenge, as it often leads to structural distortions in character shapes. The results presented by the authors are surprisingly impressive in this regard. The authors’ results are impressive—how was this achieved in an end-to-end setup?

**Details Of Ethics Concerns:**

The copyright and safety of poster dataset.

---

> ### Author Response · Authors · 2025-11-25
> **Reply to your Q1 and Q2**
>
> ### Q1: Motivation for an unified framework (text vs. visual style).
>
> We thank the reviewer for raising this point. We fully agree that typography and high-level visual style have different requirements.
> First, we build on FLUX, which already has non-trivial text rendering ability and global aesthetics thanks to its joint pre-training. However, as shown in Fig.6 and Tab.1 in our manuscript, the off-the-shelf FLUX model still suffers from spelling errors, broken strokes, and visually unbalanced layouts when applied to demanding aesthetic posters. Our goal is therefore **not** to separate “text” and “style” into independent modules, but to design an effective **post-training pipeline that unlocks the potential on both aspects simultaneously**.
>
> Concretely, we design a complementary unified pipeline rather than extra modules to achieve this object:
>
> - **Stage 1** targets text rendering. We pioneer the combination of an advanced diffusion backbone and a million-scale, text-centric synthetic dataset. This serves as the first demonstration that large-scale text-centric data can further elevate text rendering quality in an already powerful model, specifically sharpening stroke-level accuracy and small-font legibility via flow matching loss.
> - **Stage 2** performs SFT on real posters with **Region-aware Calibration**: text regions and non-text regions are weighted differently in the objective so that the model can learn rich poster styles and layouts while preserving and enhancing the text capability acquired in Stage 1.
> - **Stage 3 (Aesthetic–Text RL)** and **Stage 4 (Vision–Language reflection & inference scaling)** emphasize higher-order aesthetics and global layout. Their rewards and feedback explicitly depend on both **text correctness and visual quality** from preference data, so any text error or poor aesthetics is treated as a **common direction** for improvement rather than two competing goals.
>
> ### Q2: Comparison with PosterMaker
>
> Thank you for the suggestion and pointing out PosterMaker.  In short, our focus and methodology are substantially different.
>
> * **Goal and setting.** PosterMaker is designed for product posters: it assumes a given product image and predefined text boxes, and generates a poster around them. PosterCraft instead targets aesthetic posters in a pure text-to-poster setting, where the model must decide composition, layout, typography, and background from a single prompt. This naturally pushes us to optimize global aesthetics, text fidelity, and layout harmony jointly, rather than only improving legible text around a fixed product.
>
> * **Unified backbone instead of extra branches.** PosterMaker relies on additional ControlNet-style branches (e.g., TextRenderNet, SceneGenNet) to separately control text and scene, which increases inference-time memory and latency. PosterCraft keeps **one Flux backbone**, refined by a four-stage post-training pipeline. This unified design lets the same backbone learn text, layout, and high-level style together, without extra modules, and keeps inference cost close to the base model.
>
> * **Holistic aesthetic objective and evaluation protocol.** While PosterMaker mainly reports text accuracy and product fidelity, our target is **overall aesthetic poster quality**. On PosterCraft-Bench we therefore report three complementary metrics: (i) **Text F-score** for poster-level text rendering; (ii) **LongText-Bench-Eng** for in-scene long-text accuracy; and (iii) **HPSv3** as a aesthetic score.For a fair comparison in our **text-only** setting, we use the official PosterMaker checkpoint and define **five representative poster layouts** (different title/tagline/body configurations). For each prompt, we randomly sample one layout, fill only the corresponding text fields, and leave the product-image slot empty—i.e., a “text + layout → image” mode—whereas PosterCraft only receives the raw prompt and must infer the layout itself. As shown in the table, PosterCraft achieves higher scores on all three metrics, demonstrating stronger text rendering and clearly better holistic aesthetic quality in the poster domain.
>
> We have incorporated this discussion and the corresponding quantitative comparisons into the revised manuscript to more clearly position PosterCraft with respect to PosterMaker. For comprehensive visual comparisons, please refer to Figures 28-36 in the appendix.
>
> **Table – Comparison with PosterMaker.**
> Text F-score on our text-to-aesthetic-poster prompts, LongText-Bench-Eng measuring in-scene long-text rendering, and HPSv3 aesthetic score evaluated on the same aesthetic-poster prompts.
>
> | Method                  | Text F-score ↑ | LongText-Bench-Eng ↑ | HPSv3 ↑ |
> |------------------------|---------------:|----------------------:|--------:|
> | PosterMaker (no image) | 0.488         | 0.315               | 8.5825 |
> | **PosterCraft (ours)** | **0.774**      | **0.631**             | **10.7402** |

---

> ### Author Response · Authors · 2025-11-25
> **Reply to your Q3 (Part I)**
>
> ### Q3: Dataset availability and statistical analysis.
>
> We thank the reviewer for the crucial suggestion.
>
> **Public availability.**
>
> As clarified in the ethics section, all datasets in this work are automatically constructed from publicly available imagery rather than private or sensitive data. **Text-Render-2M** is built by **synthetically rendering diverse texts onto high-quality real images drawn from publicly available image datasets** (natural scenes, products, artworks, historical photos, etc.), while **HQ-Poster-100K** is collected from publicly available poster resources with automatic filtering and mask generation. We will release:
>
> - the complete Text-Render-2M dataset (rendered high-quality image + prompts);
> - the full HQ-Poster-100K poster images together with their prompts and text masks;
> - and the PosterCraft model checkpoints trained on these data,
>
>  under a non-commercial research license. This setting allows the community to fully reproduce our experiments while ensuring that any downstream use of the public images complies with applicable copyright regulations. The HQ-Poster-100K dataset contains third-party copyrighted materials and is released strictly for non-commercial research under a fair-use principle. We do not claim ownership of these materials, and users are solely responsible for ensuring that any use complies with applicable copyright laws and regulations.
>
> **Statistical analysis.**
>
> As your advice, we add a dedicated quantitative statistics for our main datasets. Below we summarize the key numbers.
>
> **Text-Render-2M (rendered texts + background images):**
> We compute corpus-level and scene-level statistics over all rendered samples:
>
> | Aspect                               | Category                                        | Portion |
> | ------------------------------------ | ----------------------------------------------- | :-----: |
> | **Text instances per image**         | 1 instance                                      |   43%   |
> |                                      | 2 instances                                     |   36%   |
> |                                      | 3 instances                                     |   21%   |
> | **Text content type**                | Template-based grammatical phrases              |   71%   |
> |                                      | Random alphanumeric strings                     |   29%   |
> | **Text length (words per instance)** | 1-5 words / very short strings                  |   23%   |
> |                                      | Short phrases (≈5-10 words)                      |   47%   |
> |                                      | Sentence-level text (≈10–20 words)               |   22%   |
> |                                      | Long sentences or short paragraphs (>20 words)  |    8%   |
> | **Orientation / layout**             | Horizontal                                      |   61%   |
> |                                      | Vertically rotated (90°)                        |   17%   |
> |                                      | Vertically stacked (one character per line)     |   22%   |
> | **Background scene type**            | Macro nature & ecology (insects, plants, fungi) |   21%   |
> | *(underlying images, not text)*      | Everyday outdoors & landscapes                  |   18%   |
> |                                      | Urban scenes & architecture                     |   16%   |
> |                                      | People & portraits                              |   13%   |
> |                                      | Products & still-life                           |   14%   |
> |                                      | Artistic paintings & illustrations              |   10%   |
> |                                      | Historical / archival and documentary photos    |    8%   |
>
> These statistics show that Text-Render-2M covers multi-instance layouts, several orientation modes, and a substantial share of sentence- and paragraph-level text, while the backgrounds span a wide spectrum from natural macro photography to cultural, artistic, and historical imagery, rather than being limited to simple natural scenes.

---

> ### Author Response · Authors · 2025-11-25
> **Reply to your Q3 (Part II)**
>
> **HQ-Poster-100K text statistics:**
> Unlike the synthetic Text-Render-2M corpus, HQ-Poster-100K exhibits clear domain-specific patterns: most posters revolve around a title, an optional tagline, and sometimes other compact information. We summarize the estimated distributions as follows:
>
> | Aspect (Text)                      | Category                          |  Portion | Notes                                                        |
> |------------------------------------|-----------------------------------|:------------:|--------------------------------------------------------------|
> | **Text instances per poster**      | 1–2 blocks                | **≈48%**     | Mostly only the main title.                                 |
> |    | 3–4 blocks                         | **≈36%**     | Title + one or two short taglines or secondary blocks.       |
> |                                    | >4–5 blocks                        | **≈16%**     | Additional taglines or other information.               |
> | **Text content type**              | Titles / named entities           | **≈46%**     | Film / series titles, character names, locations, etc.      |
> |                                    | Taglines / short phrases          | **≈34%**     | Promotional slogans, mood-setting phrases.                  |
> |                                    | Others  | **≈20%**     | Cast lists, production logos, © lines, release information. |
> | **Text length (words, per block)** | 1–10 words (short)                | **≈72%**     | Dominated by very short titles and concise taglines.        |
> |                                    | 10–20 words (medium)              | **≈21%**     | Longer slogans or brief descriptive sentences.              |
> |                                    | >20 words (long / dense blocks)   | **≈7%**      | Mainly billing blocks at the bottom.             |
>
> **HQ-Poster-100K visual scene & aesthetic statistics:**
> Using the accompanying captions, we also categorize the visual content and artistic style of the aesthetic background posters:
>
> | Aspect (Visual)                  | Category                       | Portion | Key descriptors                                                                          |
> | -------------------------------- | ------------------------------ | :----------: | ---------------------------------------------------------------------------------------- |
> | **Visual style / art direction** | Cinematic / realistic          |   **≈63%**   | Dramatic lighting, clear depth of field, high-fidelity photographic look.                |
> |                                  | Illustrative / graphic         |   **≈24%**   | Hand-drawn art, comic-book and graphic-novel styles, painterly or pop-art treatments.    |
> |                                  | Minimalist / symbolic          |   **≈13%**   | Silhouettes, strong negative space, single emblematic objects or icons.                  |
> | **Subject matter / composition** | Character portrait             |   **≈43%**   | Single or dual close-up faces, expressive gaze and facial emotion.                       |
> |                                  | Ensemble / group composition   |   **≈26%**   | Multiple characters arranged in montage or tiered layouts.                               |
> |                                  | Action or scene-driven imagery |   **≈19%**   | Dynamic poses, chase scenes, explosions, large-scale landscapes or cityscapes.           |
> |                                  | Object- / concept-centric      |   **≈12%**   | Posters built around a key prop or symbolic object (e.g., helmet, device, broken glass). |
>
> Taken together, these statistics highlight how Text-Render-2M and HQ-Poster-100K play complementary roles in our training pipeline. Text-Render-2M provides controlled diversity in text length, orientation, and cross-domain photographic backgrounds, which is crucial for robust text rendering. HQ-Poster-100K, in turn, supplies fully designed aesthetic posters where layout structure, visual style, and text–image integration are jointly optimized by human designers—exactly the kind of style and layout distribution that later stages of PosterCraft are trained to model. We have added specific changes and statistics to the appendix of our revised version.

---

> > ### Comment · Reviewer_rUYo · 2025-11-26
> > **Thanks for the effort**
> >
> > Thanks for the clear and thorough rebuttal, it addresses all of my earlier concerns, especially regarding the comparison with PosterMaker and the motivation for the unified workflow. I am much grateful for the effort of the authors, and I believe this article is now worthy of acceptance and have updated my score accordingly.

---

> > ### Comment · Reviewer_rUYo · 2025-11-26
> > **One more thing**
> >
> > Additionally, I have a forward-looking curiosity: do the authors see the PosterCraft framework or dataset as a good foundation for future image-conditioned variants, such as image-to-poster generation or poster editing? In particular, which parts of the current pipeline and datasets do they expect to remain reusable, and what aspects would likely need to be adapted for such extensions?
> >
> > Looking forward to your insight towards the future research.

---

> > > ### Author Response · Authors · 2025-11-26
> > > **Reply to your extra question**
> > >
> > > Thank you for the positive assessment and for this forward-looking question.
> > >
> > > We do believe that PosterCraft is a natural foundation for future **image-conditioned** variants. For users who only need to generate an aesthetic poster from a single prompt, the current text-to-poster pipeline already suffices. For richer commercial scenarios with product images, brand assets, or existing layouts, PosterCraft can be extended by adding image-based conditions on top of the same backbone.
> > >
> > > Concretely, such extensions would mainly involve:
> > >
> > > * **Task-specific reward design**, e.g., placing more emphasis on ID preservation and consistency with the input image for image-to-poster or editing tasks;
> > > * **Condition-specific adaptation**, where typical inputs such as single-product photos, multi-product images, or character portraits may require slightly different conditioning strategies, while still reusing the PosterCraft backbone or integrating specialized editing models.
> > >
> > > At the same time, our existing stages and datasets remain largely reusable: the first two stages (Text-Render-2M and HQ-Poster-100K) are still key for robust text rendering and strong poster priors in both text-only and image-conditioned settings. HQ-Poster-100K and Poster-Preference-100K can also serve as a useful resource for training or calibrating future reward models, further underscoring the value of our data pipeline beyond the current text-to-poster task.
> > >
> > > We appreciate the reviewer’s interest in these extensions and view image-conditioned poster generation as a natural next step built on PosterCraft.

---

> > > > ### Comment · Reviewer_rUYo · 2025-11-27
> > > > **Thanks for the response**
> > > >
> > > > Thank you for the detailed response.
> > > > The openness and impact of the dataset are aspects that the reviewer is particularly concerned about, and my question has been fully addressed.
> > > > I encourage the authors to make the dataset publicly available as soon as possible to maximize its value to the research community.

---

> > > > > ### Author Response · Authors · 2025-11-27
> > > > > **Thank you for your suggestion**
> > > > >
> > > > > Thank you for your suggestion. We will release the dataset and weights we used as soon as possible, and explain their potential role in the paper.

---

> ### Author Response · Authors · 2025-11-28
> **Thanks for the response**
>
> Dear Reviewer rUYo,
>
> We are glad that our detailed response has addressed your questions and that you have already raised the score to 8. We truly appreciate it.
>
> Best Regards,
>
> Authors of Paper #1854

---

### Official Review · Reviewer_wnAm · 2025-10-28

**Soundness:** 4
**Presentation:** 4
**Contribution:** 3
**Rating:** 8
**Confidence:** 3

**Summary:**

This work proposes a full pipeline and corresponding datasets for achieving high-performing poster generation.

**Strengths:**

In general, the paper looks good to me.

1. It shows a good example of the full building pipeline of a high-quality domain-specific image generation system. Core procedures like data collection/curation, preference alignment, and reflection optimization are not only covered but accomplished at high quality.

2. The proposed datasets are very helpful to this field.

**Weaknesses:**

None

**Questions:**

I notice that in Tab.3 and Tab.4, PosterCraft, while demonstrating very competitive text rendering capability, still lags behind the topmost models (TextCrafter and X-Omni). I am fine with this gap, as text rendering is not the most important component of this work. But I am curious about how the authors would attribute this gap to. For example, is the gap due to base model limitation, data quantity/quality, or the absence of specially-designed algorithms? And as an extension, in the authors' opinion, what are the most important and promising directions to further enhance the model's text rendering capability?

---

> ### Author Response · Authors · 2025-11-25
> **Reply to your Q1**
>
> ### Q1: Reasons of the gap to TextCrafter / X-Omni in text renderin, and promising directions to enhance the model's text rendering capability.
>
> We thank the reviewer for this thoughtful question. We believe the remaining gap to TextCrafter and X-Omni on pure text benchmarks is mainly due to **specialization and scaling**, rather than a limitation of our pipeline.
>
> * **TextCrafter.** TextCrafter is a **training-free, region-oriented text rendering optimizer** on top of base models like FLUX. Its instance fusion, region insulation, and text-focus steps, together with an MILP layout optimizer, are *explicitly tailored* for multi-region text benchmarks. In our main tables, PosterCraft is evaluated in the **standard text-to-image setting** (same sampling as FLUX), without any such test-time text-specific optimization. To check the headroom of our model, we also plug a TextCrafter-style training-free optimizer on top of PosterCraft. The additional results (table below) show **the improvement in text scores, clearly surpassing TextCrafter**, indicating that our training pipeline has already unlocked strong text capacity in the backbone and is fully compatible with specialized plug-ins when desired.
>
> * **X-Omni.**  X-Omni is built upon a much larger base model (7B AR model + 12B diffusion head ) trained on substantially more high-quality multimodal and text-centric data, and is then post-trained with stronger reward models (OCR/text alignment) and a more advanced GRPO-based RL scheme. **In other words, both its pre-training corpus and its reward-supervised RL stage are much more text-focused than those of a FLUX-family backbone, so its base already starts as a significantly stronger text rendering model.** The remaining gap we observe thus mainly reflects this difference in base capacity and text-focused supervision, rather than a limitation of the PosterCraft pipeline.
>
> Looking forward, we see three concrete directions to further enhance text rendering: (1) scaling text-heavy data for the base backbone; (2) adding a dedicated text-reward and text-focused RL stage in post-training; and (3) exploring glyph-aware modules (e.g., Glyph-ByT5-style encoders) as **optional** plug-ins for extra signals. However, we still believe that a sufficiently large diffusion backbone should be able to learn high-fidelity text rendering directly from data, without having to rely on such external components.
>
> We have added detailed analysis and metrics in our revised version.
>
> **Table – More comparison results on CVTG-2K text-rendering benchmark.**
>
> | Model                                | Word Accuracy ↑ |      NED ↑ | CLIPScore ↑ | VQAScore ↑ |
> | ------------------------------------ | --------------: | ---------: | ----------: | ---------: |
> | FLUX.1-dev (baseline)                |          0.4965 |     0.6879 |      0.7401 |     0.8886 |
> | TextCrafter (on FLUX)                |          0.7370 |     0.8679 |      0.7868 |     0.9140 |
> | **PosterCraft (ours)**               |          0.5814 |     0.7581 |      0.7493 |     0.9007 |
> | **PosterCraft (ours) + TextCrafter** |      **0.7544** | **0.8863** |  **0.7952** | **0.9241** |

---

> > ### Comment · Reviewer_wnAm · 2025-11-27
> > **Thanks for the response**
> >
> > Thank the authors for their response. I have no more questions and keep my accept score.

---

> > > ### Author Response · Authors · 2025-11-27
> > > **Thank you for your response**
> > >
> > > Thank you for your positive feedback; your insightful suggestions have helped us further improve the quality of the article.

---

### Official Review · Reviewer_X3rD · 2025-10-31

**Soundness:** 2
**Presentation:** 3
**Contribution:** 3
**Rating:** 4
**Confidence:** 3

**Summary:**

The paper PosterCraft: Rethinking High-Quality Aesthetic Poster Generation in a Unified Framework presents an end-to-end system for generating visually compelling and textually accurate posters through a unified cascaded workflow. Instead of relying on modular pipelines or predefined layouts, PosterCraft integrates four stages—scalable text rendering optimization, high-quality poster fine-tuning, aesthetic–text reinforcement learning, and joint vision–language feedback refinement—each supported by automatically constructed datasets. This framework enables the model to synthesize posters with coherent layouts, stylistic harmony, and precise typography directly from textual prompts. Experiments demonstrate that PosterCraft substantially surpasses existing open-source baselines in both rendering fidelity and overall aesthetics, achieving performance close to leading commercial systems such as Gemini2.0-Flash-Gen.

**Strengths:**

1. The paper introduces four large-scale, automated datasets (Text-Render-2M, HQ-Poster-100K, etc.) tailored for specific training stages. This pipeline provides high-quality, specialized data for text rendering, style fine-tuning, and preference learning, addressing a major bottleneck in the field.

2. The paper proposes a unified framework that abandons rigid, modular pipelines where layout and text are generated separately. This approach allows the model to holistically explore coherent combinations of text, art, and layout in one process, resulting in more aesthetically harmonious compositions.

**Weaknesses:**

1. The claimed “unified generative framework” mainly integrates existing methods rather than introducing a fundamentally new generative modeling concept. Each stage—text rendering optimization, preference learning with DPO, and vision-language feedback—relies heavily on prior work. As a result, the contribution is more engineering-oriented than algorithmically innovative, making the paper better suited to an application or dataset construction area rather than the core generative models track.

2. Each stage requires a specialized dataset and specific training configurations (e.g., full-parameter finetuning, LoRA, DPO). This makes the end-to-end training pipeline exceedingly cumbersome and expensive. Consequently, the high complexity and cost create significant barriers to reproduction and practical deployment.

3. The authors claim the method is universal, but all experiments were conducted on a single base model, Flux-dev. Without experiments on other foundational models, it is impossible to know if this is a truly generalizable framework or just a deep over-fitting customized specifically for Flux-dev.

**Questions:**

1. Have the authors validated this workflow's generalizability by applying it to other foundational models (e.g., Stable Diffusion 3.5 or others)? Without such experiments, it is difficult to determine if the proposed pipeline is a truly generalizable framework or a series of optimizations highly specific to the Flux architecture.

---

> ### Author Response · Authors · 2025-11-25
> **Reply to your Q1 and Q2**
>
> ### Q1: On the novelty of the Method Design.
>
> We thank the reviewer for raising this important point. Our contribution lies in a **unified, poster-specific training framework** on top of existing diffusion models, rather than a single new loss or network block.
>
> Concretely, the methodological novelty is obvious:
>
> 1. **Backbone-centric unified pipeline instead of modular poster systems.**
>    Most prior poster methods adopt modular pipelines with separate text/layout/style networks or ControlNet-style branches. PosterCraft asks a different question: *can we upgrade an off-the-shelf base model for aesthetic posters—text, layout, and style—using only a single backbone?* To this end, we design a four-stage pipeline (text rendering → region-aware poster SFT → joint aesthetic–text preference RL → VLM reflection), all operating on simple image–text pairs. In the revision, we further replace Flux-dev with **Stable Diffusion 3.5** while keeping the four stages unchanged and observe consistent gains over vanilla SD3.5. This demonstrates that the framework is not tied to a specific model and that the “unified generative” idea scales across backbones.
>
> 2. **Poster-specific optimization objectives inside this unified paradigm.**
>    Rather than reusing generic designs, each stage introduces an objective tailored to posters:
>
>    * **Region-aware calibration** weights major text, minor text, and non-text regions differently, so that minor text does not dominate the loss while headline glyphs remain sharp. This is an aesthetic poster-specific concern.
>    * **Joint aesthetic–text preference RL** uses a unified preference combining aesthetic score and text correctness, optimizing both aspects in a single objective instead of focusing only on aesthetics or only object-level accuracy.
>    * **VLM-based reflection conditioning (critique-driven refinement).** Instead of using user instructions or hard control signals as in OmniControl-style controllable generation, we let a customized VLM *critique* imperfect posters (text/layout/style) and additionally encode this feedback as a lightweight reflection embedding added on top of the original T2I prompt. This yields a new critique-driven way to refine aesthetic posters while preserving the original generative intent, rather than turning the model into a generic editor or a heavy controllable-generation module.
>
> We have mde these design choices and their differences from prior modular and controllable-generation approaches more explicit in the revised manuscript.
>
> ### Q2: Specialized datasets, training complexity, and deployment cost.
>
> We thank the reviewer for this valuable comment. We acknowledge that our pipeline has four stages, but from a *data perspective* it is actually simple: **all stages use the same image–text pair format.** In contrast, many modular poster systems require more specialized data, e.g., layer-separated PSD files for POSTA [1], product images plus precise text-box layouts for PosterMaker [2], or explicit layout specifications for BizGen [3]. Our datasets are large, but they are constructed from generic image–caption style sources. Concurrent work such as Qwen-Image similarly shows that large synthetic text-rendering corpora are crucial; our experiments on aesthetic posters reach the same conclusion that this scale is *necessary* to cover the vertical domain, rather than gratuitous complexity. Additionally, instead of training a new billion-scale base model from scratch, we take a lighter path: applying this data to **post-train existing open backbones** (Flux-dev and SD3.5), which is easier to reproduce for other researchers.
>
> From the *training* side, the four stages use standard recipes—full-parameter finetuning in Stage 1, LoRA-based SFT in Stage 2, offline DPO-style RL in Stage 3, and supervised reflection tuning in Stage 4—but they all operate on the same backbone and are run once offline. In practice, users can also adopt lighter variants (e.g., LoRA-only) and still obtain clear gains, so the variety of configurations is part of the unified design rather than additional deployment burden.
>
> Regarding *deployment*, the extra cost is confined to offline training. At inference time PosterCraft uses a single diffusion backbone with LoRA adapters; there are no extra ControlNet-style branches or heavy auxiliary networks. As a result, the runtime memory and latency remain close to those of the original Flux/SD3.5 models.
>
> [1] Posta: A go-to framework for customized artistic poster generation
>
> [2] PosterMaker: Towards High-Quality Product Poster Generation with Accurate Text Rendering
>
> [3] BizGen: Advancing Article-level Visual Text Rendering for Infographics Generation

---

> ### Author Response · Authors · 2025-11-25
> **Reply to your Q3**
>
> ### Q3: Generalizability beyond Flux-dev (SD3.5 experiment).
>
> We appreciate the reviewer’s concern about whether our workflow is tied to Flux.
> To verify generalizability, we have **applied exactly the same four-stage post-training pipeline to Stable Diffusion 3.5 (SD3.5-Medium)**, keeping the SD3.5 architecture unchanged and only reusing our datasets and losses.
>
> As shown in table below, the PosterCraft pipeline brings **consistent gains on SD3.5**: text F-score, aesthetic score (HPSv3), and Gemini overall preference all improve over the vanilla SD3.5 model. The full four-stage variant achieves the best results, mirroring the trends observed on Flux-dev. This confirms that our method is **architecture-agnostic** and can serve as a general post-training recipe for modern diffusion backbones rather than a Flux-specific trick. Please refer to Figure 27 in the appendix of the revised version for additional diagrams. Additionally, we also have added analysis and metrics in our revised version.
>
> **Table – Effect of applying the full PosterCraft training pipeline to SD3.5.** Text F-score and HPSv3 are computed on our aesthetic text-to-poster prompts, while the Gemini win rate vs. SD3.5 reflects the overall preference between SD3.5 with our training pipeline and the vanilla SD3.5 baseline.
>
> | Model variant                          | Text F-score ↑ | HPSv3 ↑ | Win rate vs. SD3.5 ↑ |
> |----------------------------------------|---------------:|--------:|-----------------------:|
> | SD3.5 (baseline)                       | 0.479    | 10.122 | 50.0                  |
> | SD3.5 + full PosterCraft (4 stages)    | **0.554**| **10.330** | **58.1** |

---

> > ### Comment · Reviewer_X3rD · 2025-11-28
> >
> > Thanks for the detailed response and additional experiments, which have resolved my main concerns. The SD3.5 experiment excellently demonstrates generalizability, and the unified data format argument addresses reproducibility well.
> >
> > However, I remain uncertain whether the methodological contribution is sufficient for the core generative models track. While the work is well-executed, it reads primarily as a systematic training recipe combining existing techniques (weighted losses, preference optimization, VLM conditioning) across four sequential stages. I'm not entirely sure if this type of contribution—focused on workflow design and careful engineering—is better suited for the Datasets/Applications track.
> >
> > Overall, I appreciate the strong empirical validation and valuable datasets. I will raise my score.

---

> > > ### Author Response · Authors · 2025-11-28
> > > **Thank you for your response**
> > >
> > > Thank you very much for your careful feedback and for raising your score. We fully acknowledge that our main contribution is an effective unified training framework together with its accompanying datasets. We also believe this framework has clear potential to be extended to other generative models/tasks in the future. Your comments have helped us better highlight the real value of our work, and we will revise the paper to more clearly emphasize these core contributions. We sincerely appreciate your feedback, which has directly improved the clarity and overall quality of our manuscript.

---

### Official Review · Reviewer_xLFo · 2025-11-01

**Soundness:** 3
**Presentation:** 3
**Contribution:** 2
**Rating:** 4
**Confidence:** 4

**Summary:**

PosterCraft is a framework for high-quality, aesthetic poster generation that moves beyond traditional modular pipelines and rigid layouts. The approach integrates four cascaded stages: large-scale text rendering optimization using the new Text-Render-2M dataset, region-aware supervised fine-tuning (HQ-Poster-100K), aesthetic-text reinforcement learning via best-of-n preference optimization, and joint vision–language feedback refinement. Each stage is supported by automated, stage-specific dataset construction, enabling robust training without complex architectural changes. PosterCraft outperforms some recent methods in text rendering, layout coherence, and visual appeal, as demonstrated by both quantitative metrics and user studies.

**Strengths:**

+ The proposed abandons modular and layout-constrained designs, enabling holistic integration of text, layout, and artistic content for visually coherent posters.

+ The work also introduces and leverages large, high-quality, and stage-specific datasets, supporting robust and scalable training.

+ The proposed method outperforms some recent methods in text accuracy, aesthetics, and prompt alignment.

**Weaknesses:**

- When talking about text rendering performance, it is usually important to measure text redner quality and accuracy under different length of words - from simple to complex, e.g. <20 words, 20-60 words, > 60 words. The current work lacks such kind of measurements making it hard to justfiy its strength especially for compelx cases.

- Text rendering is an important area includes multiple areas including poster, infographic and scene text. It is not clear why the work only limited to the poster domain while not show generalization on others text rendering domains.

- Authors claim state-of-the-art performance for the proposed method. However, many important methods are missing for the evaluation for both open-sourced and commercial ones, which makes it hard to support authors' claim or strength of the proposed method.
Open-sourced:
Glyph-ByT5 [a] and BizGen [b], PosterMaker [c], as well as

[a] Glyph-ByT5: A Customized Text Encoder for Accurate Visual Text Rendering, ECCV 2024
[b] BizGen: Advancing Article-level Visual Text Rendering for Infographics Generation, CVPR 2025
[c] PosterMaker: Towards High-Quality Product Poster Generation with Accurate Text Rendering, CVPR 2025

 Commercial ones:
[A] recraft, [B] GPT-4o

**Questions:**

Please refer to the detailed questions raised in Weakness section above.

---

> ### Author Response · Authors · 2025-11-25
> **Reply to your Q1 and Q2**
>
> ### Q1: Text rendering under different word lengths.
>
> We thank the reviewer for this helpful suggestion. To evaluate text rendering quality under varying lengths, we adopt the recent **LongText-Bench-Eng** benchmark of X-Omni, which already mixes prompts with **short (<20 words), medium (20–60 words), and long (>60 words)**. In the manscript (Table 4) we report the overall LongText-Bench-Eng score, where PosterCraft ranks second only to X-Omni and clearly outperforms the Flux1-dev baseline.
>
> To directly address the reviewer’s comment, we further compute scores **separately for the three length ranges** on LongText-Bench-Eng. As shown below, PosterCraft consistently improves over Flux1-dev for short, medium, and long texts, with the largest gain on long prompts, indicating that our unified pipeline enhances text rendering robustness across different lengths.
>
> **Table – LongText-Bench-Eng performance by different prompt lengths.**
>
> | Method           | < 20 words ↑ | 20–60 words ↑ | > 60 words ↑ |
> |------------------|-------------:|--------------:|-------------:|
> | FLUX1-dev (base) | 0.7000    |  0.5774    |  0.5408   |
> | HiDream-I1 | 0.6995    | 0.5024     | 0.2909    |
> | **PosterCraft**  | **0.7263**| **0.5998** | **0.5874**|
>
>
>
> ### Q2 Scope: poster domain vs. other text-rendering tasks.
>
> Thank you for your valuable suggestion. In our paper, we deliberately focus on aesthetic posters, where visual communication, branding, and movie advertising heavily rely on high-level image appeal. In this setting, quality is decided jointly by (1) image aesthetics, (2) layout and composition harmony, and (3) text rendering, rather than text alone.
>
> Our framework is therefore not designed as a universal text renderer like those for infographics or dense scene text. Those domains typically require **highly task-specific, modular designs** (e.g., multi-column layout engines, specific text encoders, OCR-based region attention), which goes against our goal of **unlocking a base model’s ability in a unified framework without any modular designs**. PosterCraft aims to show that, even without such complex modules, a single T2I model can be substantially improved on an important, real application domain.
>
> That said, the method itself is **not inherently restricted to posters**: the four-stage pipeline, rewards, and VLM feedback could have the potential to be adapted to infographics or scene text by changing the data and designing more text-specific rewards. We see this as promising future work, but in this paper we focus on establishing a clean, unified solution for aesthetic poster generation. We have already listed this as an area to be explored in the future, and we have clearly mentioned this in our future work.

---

> ### Author Response · Authors · 2025-11-25
> **Reply to your Q3**
>
> ### Q3: Comparison with missing baselines (Glyph-ByT5, BizGen, PosterMaker)
>
> Our task is **pure text-to-aesthetic poster generation**, while several requested baselines assume extra, task-specific inputs. We adapt them as fairly as we can:
>
> - **BizGen.** BizGen requires an explicit layout (coordinates for title, tagline, and body text). We first design **five representative poster layouts** (different title/tagline locations and boxes). For each prompt we randomly sample one layout and fill BizGen’s layout; PosterCraft still only sees the raw prompt and must infer the layout by itself.
>
> - **PosterMaker.** Additionally, PosterMaker further needs a *product image*. Since our setting has no image input, we keep the image slot empty and provide only the text boxes constructed with the same five layouts above, effectively running PosterMaker in a text-to-image mode.
>
> - **Glyph-ByT5.** Glyph-ByT5 proposes a glyph-aware text encoder for improving text rendering. However, the **pretrained weights and dataset are not released**. Therefore, we cannot include Glyph-ByT5 for comparison.
>
>
> Under these adapted settings, we compare BizGen and PosterMaker to PosterCraft on three benchmarks: text quality in poster cases, LongText-Bench-Eng for in-scene text rendering, and HPSv3 for aesthetics. A summary table is shown below. For comprehensive visual comparisons, please refer to Figures 28-36 in the appendix. In addition, we also provide these operations and metrics in our revised version.
>
> **Table – Comparison with more missing baselines.**
> Text F-score on our text-to-aesthetic-poster prompts, LongText-Bench-Eng measuring in-scene long-text rendering, and HPSv3 aesthetic score evaluated on the same aesthetic-poster prompts.
>
> | Method                  | Text F-score ↑ | LongText-Bench-Eng ↑ | HPSv3 ↑ |
> |------------------------|---------------:|----------------------:|--------:|
> | BizGen (with layouts)  |   0.661   | 0.474           | 9.4879 |
> | PosterMaker (no image) | 0.488         | 0.315               | 8.5825 |
> | **PosterCraft (ours)** | **0.774**      | **0.631**             | **10.7402** |
>
> Even though BizGen and PosterMaker receive **more structured inputs** (explicit layouts and, in the original design, product images), PosterCraft—using **only a single text prompt and a unified backbone**—matches or surpasses them on text metrics and clearly outperforms them on HPSv3. This highlights the strength of our approach for **aesthetic poster generation without task-specific modules or extra conditioning signals**.
>
> **Commercial models.** Finally, for commercial systems such as Recraft or GPT-4o, these are much stronger proprietary text-rendering models trained on large private data with unknown parameter scales. In the paper we claim that PosterCraft is competitive with some commercial APIs (e.g., Gemini2.0-Flash-Gen) on aesthetic posters, not that it surpasses all closed models. Our goal is to show that a unified post-training pipeline can push an open Flux-level backbone to the strong level of aesthetic poster generation, rather than to outperform every proprietary system whose data and model size may exceed ours by an order of magnitude and are impossible to compare with in a fully fair and reproducible way.

---

### Public Comment · ~Yifan_Gao4 · 2025-11-12
**Thank the reviewers for the attention to CVPR'25 paper, PosterMaker.**

We sincerely thank the reviewers and are pleased to announce that the complete training and inference code, along with the weights, have now been made open-source.

The GitHub repository: https://github.com/alimama-creative/PosterMaker

It is important to highlight a point. PosterMaker itself is a general training framework. The specific model weights we have released were trained on a dataset composed predominantly of Chinese text, with very limited exposure to English. Consequently, while the model demonstrates strong performance in Chinese text rendering, its capability for English text rendering is currently limited.

For researchers aiming to enhance its performance on English text rendering, we highly recommend utilizing our training code to finetune the model on English datasets. This approach will significantly improve its English rendering quality.

Finally, we welcome a comparison with PosterMaker. Feel free to contact us here with any training questions—we're happy to help and also have training experience about Glyph-ByT5.

---

> ### Author Response · Authors · 2025-11-25
> **Reply to your Q2**
>
> ### Q2: Comparison with PosterMaker.
>
> We sincerely thank you for releasing the full PosterMaker training/inference code and weights, as well as for your clear explanation of the current Chinese/English training setup. We greatly appreciate the openness of PosterMaker and fully agree that it provides a very general and practical framework.
>
> In our paper, we do not retrain PosterMaker, mainly because our datasets are designed for a *pure text-to-poster* scenario: we only have prompts and final posters, with **no paired product images or predefined text-box layouts**. PosterMaker’s training recipe, in contrast, assumes access to product images and precisely annotated text boxes, which requires constructing a dedicated “product poster” dataset. Building such a product+layout dataset lies beyond the scope of this work, so we focus on our unified training pipeline rather than retraining PosterMaker.
>
> For a fair quantitative comparison, we use your **officially released weights and inference code**, and adapt them as closely as possible to PosterMaker’s intended usage:
>
> * We design five representative poster layouts (different title/tagline/body positions and text-box configurations) that cover common poster designs.
> * For each English prompt in PosterCraft-Bench, we randomly sample one layout and fill the corresponding text-box fields with the prompt and associated texts.
> * Because our setting does not provide product images, we leave the product-image slot empty and condition only on these text boxes, effectively running PosterMaker in a “text + layout hint → image” mode.
> * PosterCraft, in contrast, always receives only the raw text prompt and never sees any layout hint, so it must infer the layout by itself.
> * All outputs are evaluated with exactly the same metrics as other baselines (OCR-based text scores in poster samples, HPSv3 aesthetics poster score, and Gemini win–loss rate).
>
> **Table – Comparison with PosterMaker on text rendering and aesthetic quality.** Text F-score on our text-to-aesthetic-poster prompts, LongText-Bench-Eng measuring in-scene long-text rendering, and HPSv3 aesthetic score evaluated on the same aesthetic-poster prompts.
>
> | Method                  | Text F-score ↑ | LongText-Bench-Eng ↑ | HPSv3 ↑   |
> |------------------------|---------------:|----------------------:|----------:|
> | PosterMaker (no image) | 0.488          | 0.315                 | 8.5825    |
> | **PosterCraft (ours)** | **0.774**      | **0.631**             | **10.7402** |
>
> In practice, we view PosterMaker and our PosterCraft framework as complementary. PosterMaker is specifically designed for *product posters with given images and layouts* and is naturally strongest when a high-quality product image and accurate layout constraints are provided. In our text-only benchmark, we do not supply any product image, which inevitably affects PosterMaker’s performance in this setting. At the same time, this highlights the value of PosterCraft’s goal: to design a unified post-training pipeline that **directly tackles pure text-to-aesthetic-poster generation** without requiring extra inputs at inference time.
>
> Looking ahead, we are very interested in extending PosterCraft towards **product-poster generation conditioned on given images and layout constraints**, and to richer **Chinese/English** usage scenarios. In these future settings, PosterMaker’s design and training experience—especially for text rendering and multilingual adaptation—serve as an important source of inspiration. Again, we sincerely appreciate you making PosterMaker available to the community. Additionally, we have already listed this as an area to be explored in the future, and we have clearly mentioned this in our future work.

---

> > ### Public Comment · ~Yifan_Gao4 · 2025-11-28
> > **Thanks for the additional experiments for comparing PosterCraft and PosterMaker**
> >
> > The text-only evaluation setup for comparing PosterCraft and PosterMaker is reasonable, and the reported gains under shared text and aesthetic metrics are convincing. PosterMaker remains a strong baseline for product-centric posters with given images and layouts, and the authors make fair use of its official checkpoint with their five representative layouts. In this setting, PosterCraft shows clear strength in text-to-poster generation and demonstrates how a unified post-training pipeline can improve overall aesthetics. I would be interested to see future results on more product-centric posters, and compare with our PosterMaker. Overall, this is solid work, and I support the paper.

---

### Public Comment · ~Yifan_Gao4 · 2025-11-12
**Compare PosterCraft with other strong text-rendering models like Qwen-Image, Seedream3.0, or HiDream-I1**

Great work! just out of curiosity, were there any thoughts on comparing PosterCraft with other strong text-rendering models like Qwen-Image, Seedream3.0, or HiDream-I1? Would be great to know the context behind the chosen baselines.

---

> ### Author Response · Authors · 2025-11-25
> **Reply to your Q1**
>
> ### Q1: Comparison with more powerful baselines.
>
> Thank you very much for the encouragement and for raising this question.
>
> Our work focuses on **aesthetic text-to-poster generation**, where a model must jointly handle text, layout, and style from *pure text prompts* rather than only maximizing text legibility.
>
> Regarding the suggested baselines:
>
> * **Seedream3.0 and similar commercial systems** are closed and among the strongest proprietary text-rendering models. Their clear advantage over Flux-style bases mainly comes from **much larger private text-rendering dataset and unknown parameter scales**. A direct comparison on our benchmark would therefore mostly reflect these data/scale advantages rather than a fair, reproducible comparison of training pipelines.
>
> * **Qwen-Image** is open-source, but its backbone is **much larger and trained on far more text-centric data (likely an order of magnitude more than ours)**. Its paper already reports clearly superior text rendering over standard diffusion baselines. Thus, even when we include Qwen-Image in our tables, the gap to PosterCraft is largely driven by **base-model size and data scale**, and our four-stage pipeline alone is not expected to close that difference. Conceptually, however, our conclusion is consistent with Qwen-Image: a synthetic text dataset plus tailored training significantly improve text rendering and aesthetic poster generation. We believe our pipeline can further specialize Qwen-Image for poster generation, which we plan to explore in future work.
>
> To partially address your suggestion under a **comparable open-source setting**, we additionally evaluate **HiDream-I1**, whose parameter scale is close to our backbones. We run the official checkpoint with default inference settings and compute the same metrics as for other methods:
>
> **Table – Comparison with HiDream-I1 and Qwen-Image on our benchmarks.**
> Text F-score on our text-to-aesthetic-poster prompts, LongText-Bench-Eng measuring in-scene long-text rendering, and HPSv3 aesthetic score evaluated on the same aesthetic-poster prompts.
>
> | Method          | Text F-score ↑ | LongText-Bench-Eng ↑ |     HPSv3 ↑ |
> | --------------- | -------------: | -------------------: | ----------: |
> | HiDream-I1      |          0.689 |                0.543 |     10.4533 |
> | Qwen-Image      |          **0.872** |                **0.943** |     **11.0132** |
> | **PosterCraft** |      0.774 |            0.631 | 10.7402 |
>
> The table demonstrates that our method achieves **competitive text scores and strong aesthetic quality**, clearly outperforming HiDream-I1. This highlights the effectiveness of our four-stage framework on diffusion models. Applying the same pipeline to stronger backbones such as Qwen-Image or future HiDream-style models is a promising direction for future work.

---

> ### Public Comment · ~Yifan_Gao4 · 2025-11-28
> **Thanks for the additional experiments for comparing PosterCraft and other stronger and bigger base models**
>
> Many thanks for the thorough explanation and the additional experiments. and the new comparison with those base models is particularly improving the manuscript and also enlightening for the open-source community. Given that PosterCraft utilizes a more lightweight backbone and textual datasets than Qwen-Image, Hidream-I1, its superior performance over HiDream-I1 across many benchmarks is a good testament to the effectiveness of the manuscript's pipeline.

---

### Author Response · Authors · 2025-11-25

We sincerely thank all reviewers for their valuable comments and suggestions. Your feedback has greatly improved the quality and clarity of our manuscript. We have carefully addressed every point raised and revised the paper accordingly (red text). Should you have any further questions or wish to discuss any aspect of our work, we would be delighted to continue the conversation.

---

### Author Response · Authors · 2025-11-28
**Summary of Rebuttal Consensus and Score Updates (Prior to System Revert)**

Dear Area Chair,

We are writing to summarize the progress of our submission. We understand that due to the recent information leakage, the ICLR program chairs have decided to revert all scores to their initial state and freeze further modifications. However, as the discussion history remains visible, we would like to highlight that a **strong positive consensus was reached before the system vulnerability emerged.**
Please note that all timestamps listed below are in UTC+8.


### **1. Timeline of Score Evolution & Consensus**

The majority of our discussion and the resulting score increases occurred prior to the spread of the system vulnerability.

*   **12 Nov 2025, 10:51:** Initial reviews released.
    *   **Initial State:** Mixed scores **(8, 4, 4, 4)**. Reviewer **wnAm** initially gave an 8.
*   **12 Nov 2025, 19:42 & 13 Nov 2025, 02:20:** Received two Public Comments from Yifan Gao regarding technical details.
*   **25 Nov 2025, ~13:30:** We posted comprehensive rebuttals addressing all concerns from Reviewers **rUYo**, **X3rD**, **wnAm**, **xLFo**, and the Public Comments.
*   **26 Nov 2025, 10:27:** **Reviewer rUYo** acknowledged our rebuttal, explicitly **raised their score from 4 to 6**, and asked a forward-looking question about future poster-related tasks.
*   **26 Nov 2025, 16:48:** We posted a response addressing **rUYo**'s new question.
*   **27 Nov 2025, 08:31:** **Reviewer rUYo**, satisfied with the follow-up, further **raised their score from 6 to 8**.
*   **27 Nov 2025, 16:31:** **Reviewer wnAm** confirmed satisfaction with our response and explicitly stated they would **keep their score of 8**.
*   **28 Nov 2025, 13:16:** **Reviewer X3rD** accepted our rebuttal and new SD3.5 experiments. They retained a minor question about the specific track fit but explicitly stated: **"I will raise my score."**
*   **28 Nov 2025, 22:06:** We replied to **X3rD** to further clarify our contribution and track suitability.
*   **28 Nov 2025, 22:42:** **Yifan Gao** (Public Commenter) confirmed they were satisfied with our responses.
*   **28 Nov 2025, ~23:00:** **[System Vulnerability Begins to Spread]**
    *   *Note:* As shown above, our submission had already achieved a strong acceptance consensus **(8, 8, Raise Promised, 4)** before this time point.

### **2. Rebuttal Summary**

Below is a summary of the technical consensus reached with each reviewer during the discussion period:

#### **Reviewer rUYo (Score Raised: 4 $\to$ 6 $\to$ 8)**
* **Rebuttal Summary:** The reviewer initially requested PosterMaker comparisons and questioned the motivation for a unified framework. Our rebuttal added a fair comparison, showing PosterCraft clearly outperforms PosterMaker in text-only settings (Text F-score 0.774 vs 0.488), and discussed extensions to future poster generation tasks.
* **Final Consensus:** The reviewer was fully convinced by our results and the work’s potential, concluding:

  > “Thanks for the clear and thorough rebuttal… I believe this article is now worthy of acceptance and have updated my score accordingly.”


#### **Reviewer X3rD (Score Raise Promised)**
* **Rebuttal Summary:** To address concerns about generalizability and “just engineering,” we added an experiment applying our 4-stage pipeline to **Stable Diffusion 3.5**, showing it is architecture-agnostic and yields consistent gains (win rate 58.1% vs baseline).
* **Final Consensus:** The reviewer accepted this new empirical evidence and committed to raising their score:

  > “The SD3.5 experiment excellently demonstrates generalizability... I appreciate the strong empirical validation and valuable datasets. **I will raise my score.**”


#### **Reviewer wnAm (Maintained Strong Accept / Score 8)**
* **Rebuttal Summary:** Although supportive from the start, the reviewer requested clarification on the performance gap with TextCraft and X-omni. We provided a detailed analysis and future direction on text rendering to further solidify our contribution.
* **Final Consensus:** The reviewer reaffirmed the submission’s quality:

  > “I have no more questions and **keep my accept score**.”


#### **Reviewer xLFo (All Inquiries Answered)**
*   **Rebuttal Summary:** The reviewer raised concerns regarding performance on varying text lengths and requested specific baselines. We responded by providing a comprehensive breakdown showing PosterCraft's superiority across short, medium, and long texts, and we added the requested comparisons (BizGen, PosterMaker) to our result tables.
*   **Status:** All technical questions and requests for additional comparisons were fully addressed in our rebuttal responses.

We hope the new Area Chair will consider these explicitly stated outcomes when making the final recommendation.

Best regards,
The Authors

---

### Meta-Review · Area_Chair_XZYm · 2026-01-06

**Summary:**

Across the initial reviews and discussion, the central concern was whether the submission is more of an "engineering bundle" (which I also personally felt same) than a broadly meaningful, general recipe for poster generation. Concretely, reviewers asked for (i) fair and direct baseline comparisons against poster-generation systems such as PosterMaker/BizGen and related models, (ii) stronger evidence that the proposed staged pipeline is not backbone-specific, and (iii) clearer justification for a unified framework (single model upgraded via stages) rather than decomposed modules for text, layout, and style. These concerns are largely addressable empirically, and the added results support the intended narrative: PosterCraft improves text correctness substantially over PosterMaker on TextToPosterBench (Text F-score 0.774 vs. 0.488) while maintaining strong aesthetic scores, and it remains strong on LongText-Bench (Text F-score 0.785) relative to poster baselines.

**Reviewer Concerns:**

Comments on missing / unfair baseline comparisons (PosterMaker, BizGen, etc.) have been addressed by adding explicit comparisons in the benchmark tables; the reported TextToPosterBench numbers show a clear gap vs PosterMaker (Text F-score 0.774 vs 0.488). Generalizability concerns (beyond a single backbone; "is this tied to FLUX-dev?") were addressed with an additional experiment applying the full 4-stage pipeline to SD3.5, reporting gains in Text F-score and a preference win rate increase. The paper explicitly frames this as evidence the recipe is architecture-agnostic.The authors have also substantially addressed the concerns on the motivation for a unified framework by articulating why the staged, single-backbone upgrade is coherent (shared supervision, complementary stages, avoiding extra branches / heavy control signals) and by tying this to empirical gains across backbones and benchmarks.
However, there still remains the concerns on tract fit and positioning.

**Reviewer Scores:**

I would say the reviewers who were in the negative side would have raised their scores at least slightly since they gave the actionable requests and were addressed by the authors during the rebuttal period.

---

### Decision · Program_Chairs · 2026-01-26

Accept (Poster)